# DualMap: Enabling Both Cache Affinity and Load Balancing for Distributed LLM Serving

**Ying Yuan**[*]
Huazhong University of Science and Technology

**Pengfei Zuo**[†]
Huawei

**Bo Wang**
Huawei

**Zhangyu Chen**
Huawei

**Zhipeng Tan**[†]
Huazhong University of Science and Technology

**Zhou Yu**
Huawei

## Abstract

In large language model (LLM) serving, reusing the key-value (KV) cache of prompts across requests is a key technique for reducing time-to-first-token (TTFT) and lowering serving costs. Cache-affinity scheduling, which co-locates requests with the same prompt prefix to maximize KV cache reuse, often conflicts with load-balancing scheduling, which aims to distribute requests evenly across compute instances. Existing schedulers struggle to reconcile this trade-off, as they operate within a single mapping space, typically applying cache-affinity routing to a subset of requests and load-balanced routing to the rest, without a unified solution to achieve both goals. To overcome this limitation, we propose *DualMap*, a dual-mapping scheduling strategy for distributed LLM serving that simultaneously enables cache affinity and load balancing. The key idea of *DualMap* is to map each request to two candidate instances using two independent hash functions based on the request prompt, and then intelligently select the better candidate based on current system states. This design increases the likelihood that requests with shared prefixes are co-located, while evenly dispersing distinct prefixes across the cluster via "the power of two choices". To make *DualMap* robust under dynamic and skewed real-world workloads, we incorporate three techniques: 1) *SLO-aware request routing*, which prioritizes cache affinity but switches to load-aware scheduling when TTFT exceeds the SLO, enhancing load balance without sacrificing cache reuse; 2) *hotspot-aware rebalancing*, which dynamically migrates requests from overloaded to underloaded instances, mitigating hotspots and rebalancing the system; 3) *lightweight dual-hash-ring scaling*, which leverages a dual-hash-ring mapping to support fast and low-overhead instance scaling without costly global remapping. Experiments on real-world workloads show that *DualMap* improves effective request capacity by up to 2.25× under the same TTFT SLO constraints, compared with the state-of-the-art work. Code can be found at: `https://github.com/ASISys/DualMap`.

## 1 Introduction

Recently, large language models (LLMs) have exhibited strong performance in scenarios such as multi-turn conversations and agent-based applications (Sha, 2025; Gao et al., 2024; Hao et al., 2023). A common characteristic of these scenarios is the repeated use of prompt prefixes. For example, in conversational sessions, requests share the dialogue history, while agent tool usage typically includes repeated instruction prompts.

To reduce redundant prompt computation, modern LLM serving systems widely adopt context caching—also referred to as prompt caching or prefix caching—which stores the historical key-value (KV) cache of prompts (Gao et al., 2024; Qin et al., 2025; Dyn, 2025; 202, 2025; Zheng

---

[*]Work done during her internship at Huawei.

[†]Corresponding authors are Pengfei Zuo (pfzuo.cs@gmail.com) and Zhipeng Tan (tanzhipeng@hust.edu.cn).

et al., 2024a). This allows subsequent requests to directly reuse the stored KV cache, eliminating the need for repeated prefill computation. This technique significantly improves GPU utilization and reduces inference latency, especially the time-to-first-token (TTFT)—the latency from request arrival to the first token—which is critical to user experience. Ensuring TTFT stays within a service level objective (SLO), is essential in practice. Under such constraints, system efficiency is commonly measured by *effective request capacity*, defined as the proportion of served requests that meet the target SLO (Qin et al., 2025).

In production environments, LLMs are deployed in distributed serving clusters, where request scheduling plays a crucial role in reducing inference latency and cost (Qin et al., 2025; Dyn, 2025; Srivatsa et al., 2024; Cao et al., 2025). Effective scheduling must ensure both *cache affinity*, which routes requests to compute nodes that store the KV cache of their prompt prefixes to maximize cache reuse and minimize TTFT, and *load balancing*, which distributes requests evenly across nodes to avoid hotspots and improve resource utilization.

However, achieving cache affinity and load balancing often conflict with each other. Specifically, a *cache-affinity* strategy maps requests with identical prompt prefixes to the same instance using a prompt-aware function. While this approach maximizes KV cache reuse, it does not guarantee load balancing. This is because prompt popularity in real-world workloads is typically skewed (Wang et al., 2025b), causing nodes responsible for hot prefixes to become overloaded, while those handling cold prefixes remain underutilized. In contrast, load-balancing strategies such as the *Least Loaded* strategy map requests to the instance with the lowest load, without considering KV cache hits. While this approach achieves even load distribution, it scatters requests with shared prefixes across different instances. This reduces the KV cache hit rate and increases recomputation overhead.

The fundamental trade-off between cache affinity and load balancing arises because both cache-affinity and load-balancing strategies are implemented within *a single mapping space*. *Cache-affinity* strategies rely on a prompt-aware mapping function that prioritizes routing requests to nodes with relevant KV caches, while load-balancing strategies use a load-aware mapping function to evenly distribute requests across nodes. Prior work, such as *Mooncake* (Qin et al., 2025), *Preble* (Srivatsa et al., 2024), and *Dynamo* (Dyn, 2025), attempts to balance cache affinity and load balancing, but fails to achieve both simultaneously. This is because they remain constrained by a single mapping space, typically applying prompt-aware routing to a subset of requests and load-aware routing to the rest.

To break the trade-off between cache affinity and load balancing, we propose *DualMap*, *a dual-mapping scheduling strategy* for distributed LLM serving that enables both objectives simultaneously. *DualMap* draws inspiration from the *power of two choices (PoTC)* principle (Mitzenmacher, 2002), which demonstrates that selecting the less loaded option among two randomly chosen candidates can significantly improve the system load balancing. Specifically, *DualMap* employs two independent hash functions, $f_1$ and $f_2$, to map each request to two candidate compute nodes, and dispatches the request to the more suitable node based on the current system state. The randomness of the two hash functions ensures requests are distributed evenly across the cluster. To enable cache affinity, *DualMap* uses a request's partial prefix as the input key to the two hash functions. This increases the likelihood that requests with shared prefixes are consistently mapped to the same node, where their KV caches can be stored and reused locally. In this way, *DualMap* simultaneously promotes cache affinity and load balancing.

However, implementing this design in real-world serving systems introduces three challenges:

***1) How to select the optimal one from two candidate instances to maximize system effective request capacity?*** Selecting between two candidate instances also involves a trade-off between cache affinity and load balancing. ***2) Skewed prefix popularity leads to hotspot instances.*** The PoTC principle achieves load balancing under the assumption that requests are randomly distributed across all instances. In practice, this assumption is often violated due to skewed prompt popularity. For example, frequently invoked tools can cause a large number of requests with identical prefixes to concentrate on a small subset of instances (Wang et al., 2025b)(detailed in §A.2.2). This leads to load imbalance and high TTFT tail latency. ***3) Static hash mappings limit system elasticity.*** Static hash mappings tightly bind request prefixes to specific instances. Scaling operations (e.g., adding or removing instances) disrupt these mappings, degrade cache hit rates, increase recomputation, and introduce service jitter, ultimately compromising elasticity.

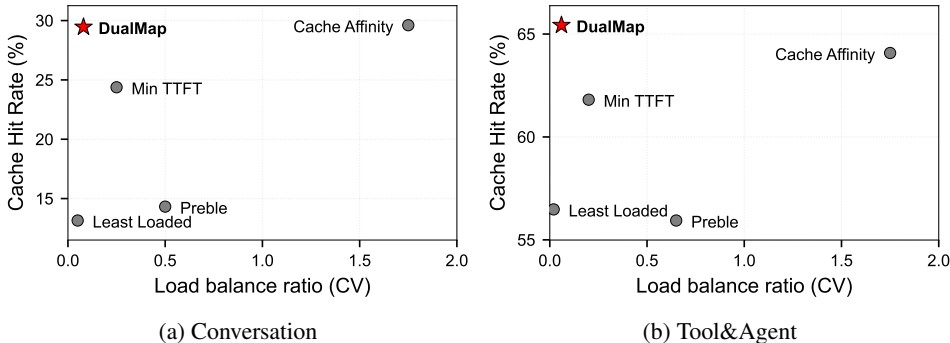

Figure 1: Pareto trade-off between cache hit rate and load balance ratio (coefficient of variation, CV) across different scheduling strategies on the *Conversation* and *Tool&Agent* datasets. A lower CV indicates more even load distribution across instances.

To tackle these challenges, *DualMap* incorporates three key techniques. First, to select the optimal instance from two candidates for maximizing system effective request capacity, *DualMap* introduces *an SLO-aware routing strategy*, which prioritizes prompt-aware scheduling to achieve cache affinity and minimize recomputation overhead, but dynamically shifts to load-aware scheduling only when expected TTFT exceeds the predefined SLO, ensuring stable performance under fluctuating load.

Second, to resolve hotspots caused by skewed prefix popularity, *DualMap* introduces *a hotspot-aware rebalancing strategy*. It selectively migrates requests from overloaded instances to their backup instances, i.e., the alternative instance from the initial dual mapping. *DualMap* prioritizes migrating the requests whose backup instance is underutilized and exhibits high cache reuse, enabling effective load redistribution without significantly compromising cache affinity.

Third, to support efficient elasticity, *DualMap* adopts *a dual-hash-ring scaling strategy*. Request-to-instance assignments depend solely on the relative positions in the hash rings, ensuring that scaling operations (e.g., instance addition or removal) affect only a small portion of the mappings. This design avoids expensive global remapping and enables fast adaptation to dynamic workloads.

We implement *DualMap* in a distributed LLM serving system with the vLLM engine (Kwon et al., 2023). Experimental results on real-world workloads demonstrate that *DualMap* improves effective request capacity by 2.25× compared to state-of-the-art scheduling approaches.

## 2 BACKGROUND AND MOTIVATION

### 2.1 LLM INFERENCE AND CONTEXT CACHING

Transformer-based LLM inference first processes the input prompt in parallel to produce the initial output and KV caches, which are then used for autoregressive decoding (Pope et al., 2023; Kwon et al., 2023; Zheng et al., 2024a). The KV caches are often stored and reused across requests, a technique known as context caching, prompt caching, or prefix caching (Gao et al., 2024; Srivatsa et al., 2024; Qin et al., 2025; Dyn, 2025). KV caches for a shared prefix $(x_1, \ldots, x_k)$ are identical and can be reused across requests, avoiding redundant prefill computation. Prefix sharing, where multiple requests share a common prefix, frequently arises in practical scenarios such as multi-turn conversations and tool-agent interactions (Sha, 2025; Qin et al., 2025). Reusing KV caches for these shared prefixes instead of recomputing them can significantly reduce prefill latency and improve inference efficiency (Gao et al., 2024; Srivatsa et al., 2024; Qin et al., 2025; Dyn, 2025).

### 2.2 SCHEDULING IN DISTRIBUTED LLM SERVING

Ideally, an efficient scheduler should achieve both high cache efficiency and balanced load. However, these goals are inherently conflicting.

**Conflicts between Cache Affinity and Load Balancing.** To systematically examine the impact of cache affinity and load balancing on system, we evaluate *Cache Affinity* and *Least Loaded* strategies on the *Conversation* (Qin et al., 2025) and *Tool&Agent* (Qin et al., 2025) datasets using an 8-instance cluster serving the Qwen2.5-7B model (Team, 2024). Detailed experimental settings are provided in §4. As shown in Figures 1a and 1b, the *Cache Affinity* strategy achieves a cache hit rate $1.21\times$ higher than that of *Least Loaded* on the *Conversation* dataset. On the *Tool&Agent* dataset, *Cache Affinity* approaches the theoretical upper bound, while *Least Loaded* falls close to the lower bound. This is because *Cache Affinity* dispatches requests with shared prefixes to the same instance, preserving cache locality and maximizing KV cache reuse. In contrast, *Least Loaded* considers only the current load of each instance (e.g., pending tokens for prefill), ignoring whether the required KV cache is already present. As a result, it scatters requests and significantly reduces cache hit rates.

To evaluate how well each strategy balances load, we measure the *coefficient of variation (CV)*, a standard metric that quantifies the degree of load imbalance across instances:

$$\text{CV} = \frac{\sqrt{\frac{1}{n}\sum_{i=1}^{n}(x_i - \mu)^2}}{\mu} \tag{1}$$

Here, $x_i$ denotes the number of pending prefill tokens on instance $i$, $\mu = \frac{1}{n}\sum_{i=1}^{n} x_i$ represents the average number of pending prefill tokens across all $n$ instances in the cluster. A lower CV indicates better balance, with a CV of zero meaning perfectly even load. As shown in Figures 1a and 1b, *Least Loaded* achieves near-zero CV, indicating highly uniform load distribution. In contrast, *Cache Affinity* results in significant load imbalance. This is because when a particular prefix becomes highly popular, all associated requests are directed to a single instance, causing queue buildup, elevated tail TTFT, and underutilization of the remaining instances.

Co-locating requests improves cache reuse, while spreading them evenly improves load balance and resource utilization. Within a unified scheduling space, optimizing for one typically comes at the cost of the other.

**Existing Trade-off Approaches.** Recent approaches (Srivatsa et al., 2024; Dyn, 2025; Qin et al., 2025) strive to balance cache affinity and load balancing through unified scheduling mechanisms. However, they fundamentally fall short of achieving both objectives simultaneously, as improvements in one often come at the expense of the other. Specifically, *Preble* enables prompt-aware scheduling when a request's prefix hit rate exceeds 50%, but switches to load-aware scheduling otherwise. Similarly, *Dynamo* and *Mooncake* adopt prompt-aware scheduling under cluster load imbalance, and switch to load-aware scheduling when the load is balanced, detailed in §B

In the evaluation, we compare only with Mooncake's request scheduling strategy, simplify *Mooncake* to *Min TTFT*, and exclude *Dynamo* since its cost-based design is similar to that of *Mooncake*. As shown in Figures 1a and 1b, the *Min TTFT* and *Preble* achieve cache hit rates and load balancing ratios between those of *Cache Affinity* and *Least Loaded*, indicating that they cannot simultaneously achieve both cache affinity and load balancing, because they apply a prompt-aware mapping to a subset of requests and a load-aware mapping to the rest.

### 2.3 MOTIVATION

As stated in §1, to overcome the limitation that a single mapping space cannot simultaneously guarantee both cache affinity and load balancing, we propose a dual-mapping scheduling approach for distributed LLM serving, inspired by the PoTC principle.

**Cache Affinity Guarantee.** For $m$ requests with the same prompt prefix $p$, *DualMap* consistently maps them to the same candidate instance set $\{I_1, I_2\}$, guaranteeing a cache hit rate of $\max(0, 1 - 2/m)$. In contrast, the *Cache Affinity* strategy achieves a cache hit rate of $\max(0, 1 - 1/m)$. When $m$ is large, *DualMap* approaches the cache hit rate of *Cache Affinity*.

**Load Balancing Guarantee.** *DualMap* adopts the PoTC strategy (Mitzenmacher, 2002), a classic scheduling paradigm widely used for its strong theoretical guarantees on load balancing in large-scale systems. For $m$ incoming requests distributed across $n$ instances, PoTC ensures that the maximum load remains tightly concentrated around the mean. When each request is mapped

to $d$ candidate instances chosen uniformly at random, the maximum load satisfies the following bound (Mitzenmacher, 2002):

$$\max_i L(I_i) \leq \frac{m}{n} + \frac{\log \log n}{\log d} + \mathcal{O}(1), \tag{2}$$

where $L(I_i)$ denotes the number of requests assigned to instance $I_i$, and $\frac{m}{n}$ is the ideal average load per instance. The second term, $\frac{\log \log n}{\log d}$, captures the deviation from the average load caused by randomness.

*Single-choice ($d = 1$).* The above bound degenerates to a much weaker guarantee:

$$\max_i L(I_i) = \frac{m}{n} + \Theta\left(\sqrt{\frac{m \log n}{n}}\right), \tag{3}$$

indicating a significantly larger deviation from the mean, which grows with both $m$ and $n$.

*Two-choices ($d = 2$).* The deviation term becomes $\log \log n$, which is exponentially smaller than that under $d = 1$. In particular, for $d = 2$, PoTC yields:

$$\max_i L(I_i) \leq \frac{m}{n} + \log \log n + \mathcal{O}(1), \tag{4}$$

leading to exponentially better load balancing than single-choice. Consequently, *DualMap* deliberately chooses $d = 2$, providing significantly tighter bounds on load imbalance, especially in large-scale deployments.

**Why Two Choices Instead of More.** Intuitively, using more choices should lead to better load balancing. However, according to Eq. 2, although increasing the number of choices $d$ reduces the deviation term $\frac{\log \log n}{\log d}$, the improvement quickly exhibits diminishing returns. For instance, the reduction from $d = 2$ to $d = 3$ or $d = 4$ is marginal (detailed in §A.9). In contrast, increasing $d$ expands the candidate instance set for each request, dispersing requests with the same prefix across more instances and thereby weakening KV cache locality. A global strategy that "collects information from all instances and selects the best one" is effectively equivalent to using $d = n$ choices. Yet Eq. 2 shows that this yields negligible improvement over $d = 2$ in terms of maximum load, while severely degrading KV cache reuse: prefix-sharing requests may be scattered across the entire cluster, increasing prefill computation and ultimately worsening TTFT. Therefore, *DualMap* deliberately adopts $d = 2$, achieving strong load balancing while preserving KV cache locality.

## 3 THE DUALMAP DESIGN

### 3.1 OVERVIEW

To break the trade-off between cache affinity and load balancing, we propose *DualMap*, a dual-mapping scheduling strategy for distributed LLM serving that enables both objectives simultaneously. Figure 2 illustrates the system architecture of *DualMap*, comprising two main components: the *global scheduler* and the *inference cluster*. Each inference instance hosts an LLM and is equipped with a given-size host DRAM used for context caching, enabling KV cache reuse across requests to reduce redundant computation and improve serving efficiency. The global scheduler handles incoming requests from external users or client applications and dispatches each request to an appropriate inference instance based on *DualMap*. Upon receiving a request, *DualMap* employs the following three techniques to achieve both cache affinity and load balancing: 1) *SLO-aware Request Routing(§3.2)* selects the most suitable instance among two candidates to maximize system SLO compliance; 2) *Hotspot-Aware Rebalancing(§3.3)* rebalances hotspot instances to achieve load balance under skewed workloads; 3) *Lightweight Dual-hash-ring Scaling(§3.4)* enables rapid scaling and resizing of the cluster.

### 3.2 SLO-AWARE REQUEST ROUTING

To balance cache reuse and load distribution, the *DualMap* global scheduler maps each request to two candidate instances using two independent hash functions over the request's prompt prefix. This process must answer two key questions:

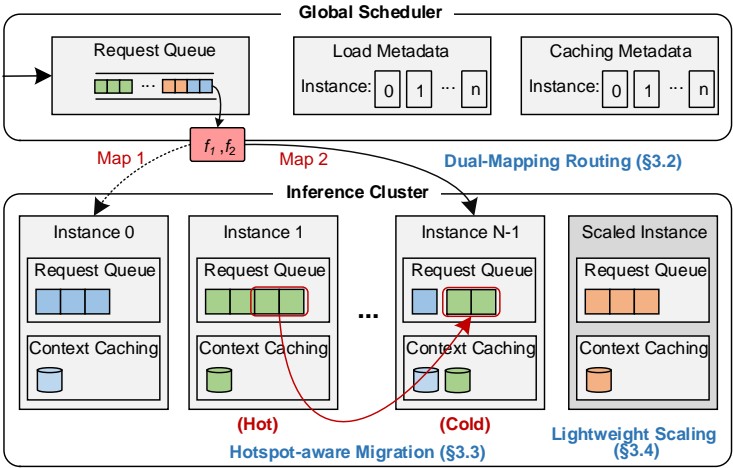

Figure 2: The system overview of *DualMap*.

**1) How long prefix is appropriate for the hash key?** Ideally, the hash key should consist of the shared prefix so that requests with shared context are mapped to the same instance, enabling KV cache reuse. In practice, however, the global scheduler does not know the shared prefix length beforehand, and different requests may have varying shared prefix lengths.

Two issues arise: (1) if the hash key is too long, it may exceed the actual shared prefix, causing requests to be mapped to different instances and reducing cache reuse; (2) if the hash key is too short, different request sets may collide, concentrating load on a few instances and causing severe imbalance. Real-world request patterns are also highly skewed, with some prefixes appearing disproportionately and forming hotspot instances.

To address this, we propose an adaptive hash prefix length mechanism that dynamically determines the prefix length for each request. The global scheduler maintains a request prefix hotness tree, where each node records prefix information. The hash prefix corresponds to the full path from root to leaf. When a leaf prefix becomes hot, child nodes are added to extend the hash prefix, distributing hot requests across more instances; when a parent prefix becomes cold, its child nodes are removed to shorten the prefix, aggregating normal requests to the same instance and improving cache reuse. Prefix hotness is tracked via the traffic ratio $\rho$ of each prefix in real time using a sliding window. Here, $\rho$ is defined as the fraction of total requests within the window that share the same prefix. A prefix is *hot* if $\rho > \frac{2}{n}$, where $n$ is the number of inference instances, reflecting the dual-mapping strategy's upper bound. If a hot prefix's traffic drops significantly between windows (e.g., $\rho > \frac{2}{n}$ to $\rho < \frac{1}{n}$), its hotness is updated and child nodes removed to shorten the prefix.

In experiments, the distribution of hash key prefix lengths for the *Conversation* and *Tool&Agent* datasets is detailed in §A.2.1.

**2) Select which intance between the two candidates?** Selecting between the two candidates involves a critical trade-off between cache affinity and load balancing, which directly impacts the request's TTFT. A more detailed analysis is provided in the §A.2.1

**Baseline Strategies.** A naive least-loaded strategy selects the instance with the lower load, disregarding cache availability. In contrast, a pure cache-affinity strategy always routes a request to the instance with the highest expected cache reuse. While this minimizes recomputation, it can lead to severe load imbalance. To minimize the risk of TTFT SLO violations, the *Min TTFT* policy estimates TTFT for each candidate instance and selects the one with the lower value. For a request $r$ and candidate instance $i$, we define: $T_q(r, i)$ as estimated queuing delay of Request $r$ on Instance $i$, capturing system load, $T_c(r, i)$ as expected computation time for Request $r$ on Instance $i$, depending on whether cache reuse occurs. Thus, the estimated total TTFT is $TTFT(r, i) = T_q(r, i) + T_c(r, i)$. However, as new requests arrive, this strategy may oscillate between cache-aware and load-aware decisions, leading to frequent cache misses under load fluctuation, increasing recomputation overhead

and worsening overall TTFT. Fundamentally, Min TTFT's pursuit of optimal latency per request inadvertently degrades cache reuse and destabilizes load balance over time.

**SLO-Aware Strategy.** To address these issues, *DualMap* adopts a SLO-aware routing strategy that explicitly incorporates TTFT constraints. The key idea is to maintain cache affinity whenever possible, and only trade it off when load imbalance threatens to breach SLO constraints. *DualMap* prioritizes cache affinity and initially selects the instance with the highest cache reuse. Requests are routed to this instance until its load causes the expected TTFT to exceed the SLO. At that point, *DualMap* switches to a load-aware strategy and selects the less loaded instance to avoid long queues and reduce TTFT tail latency. Furthermore, if both instances have equal prefix hit rates, *DualMap* always chooses the less-loaded one, further enhancing load balance without sacrificing reuse. Unlike *Min TTFT*, *DualMap* does not seek per-request optimal TTFT. Instead, it preserves cache reuse whenever load conditions permit, switching to load balancing only when necessary. This reduces the frequency of recomputation and stabilizes cache hit rates.

To quantify system load, *DualMap* uses the number of pending prefill tokens on each instance, as prefill computation latency scales with token count. Given a predefined TTFT SLO, we compute the maximum number of pending prefill tokens that a GPU can process within the SLO, referred to as the *ttft_slo_threshold*. This threshold is used as the switching criterion to a load-aware strategy in our implementation, ensuring that routing decisions remain aligned with service-level constraints.

*DualMap* is designed so that every request has two distinct candidate instances. If the two hash functions $f_1$ and $f_2$ initially map a request to the same instance, we deterministically adjust the second candidate as

$$\text{instance\_id}_2 = (\text{instance\_id}_1 + 1) \bmod \text{num\_instances}. \tag{5}$$

This ensures that each request always has two distinct candidate instances.

### 3.3 HOTSPOT-AWARE REQUEST REBALANCING

As discussed in Challenge 2 (§1), for the skewed nature of real-world request patterns, some instances may become overloaded over time—resulting in long queues and elevated tail TTFT, while others remain underutilized.

To solve this problem, we introduce *hotspot-aware request rebalancing*, selectively migrating pending requests from overloaded instances to less-loaded ones. This rebalancing strategy is inspired by Cuckoo hashing (Pagh & Rodler, 2001), where each key has two possible slots and can be relocated if its primary slot is full. In this setting, a *DualMap* instances play the role of slots, and requests act as keys. When an instance becomes overloaded (i.e., has a long queue of pending requests), some requests are redirected to their alternative candidate instance, preserving the mapping consistency of *DualMap*. Unlike traditional Cuckoo hashing, which may involve recursive evictions, we adopt a non-recursive, single-round batch migration to minimize overhead. Specifically, we evaluate multiple queued requests on overloaded instances and migrate those whose alternative instance is underloaded and yields a net TTFT benefit.

Request migration directly impacts TTFT. A straightforward approach is to migrate requests near the tail of the queue, as they experience long queuing delays. However, if their alternative instance is also congested, such migration may worsen overall latency. Another heuristic is to migrate requests with low cache affinity to reduce potential cache hit loss, but this too can backfire if the target instance has even less relevant cached data. To balance these trade-offs, we estimate the potential TTFT gain for each request by jointly considering queuing delays and computation costs on both the source and target instances. We define the *migration benefit* for a Request $r$ currently queued at the overloaded Instance $i$ with the alternative Instance $j$ as:

$$\text{B}_r^{(i \to j)} = TTFT_{r,i} - TTFT_{r,j} = (T_q(r,i) + T_c(r,i)) - (T_q(r,j) + T_c(r,j)) \tag{6}$$

Only requests with a sufficiently positive $\text{B}_r^{(i \to j)}$ are considered for migration. Requests are prioritized by descending benefit and migrated until all requests in the overloaded instance's queue are expected to meet the TTFT SLO. The alternative instance $j$ is precisely the second candidate from the dual mapping. We do not search over all instances; instead, we preserve the prefix-bound candidate pair $\{I_1, I_2\}$ and migrate requests only within this pair. This approach maintains cache affinity while keeping scheduling complexity under control. Overall, this hotspot-aware migration

mechanism effectively alleviates overloaded instances while preserving cache affinity. Details are in §A.2.2.

### 3.4 LIGHTWEIGHT DUAL-HASH-RING SCALING

LLM inference workloads often experience pronounced temporal variation in request volume. To achieve low queuing latency and high resource utilization, the system must support dynamic and responsive elastic scaling. However, static hash-based mapping poses significant challenges: scaling operations such as adding or removing instances trigger global remapping, which severely disrupts cache affinity.

To enable elastic scaling with minimal disruption, we adopt *a dual-mapping hash ring* that combines dual mapping with consistent hashing (Karger et al., 1997) to achieve lightweight elasticity. The hash ring spans a logical space $[0, M)$, with each instance assigned an anchor point based on a unique identifier (e.g., IP and port). Each request hashes its prefix using two independent hash functions to generate two anchor points and selects the nearest clockwise instance as a candidate. The scheduler chooses between these two candidates based on the SLO-aware request routing (§3.2). Because request-to-instance mappings are determined by relative positions on the ring, changes to cluster membership affect only localized regions. This design ensures that most requests retain their original mapping paths during scaling operations. As a result, cache loss is significantly reduced and elastic scaling is achieved with minimal disruption.

## 4 PERFORMANCE EVALUATION

### 4.1 EXPERIMENTAL SETUP

**Testbed.** We implement DualMap as an independent global scheduling layer for distributed LLM serving systems. In our experiments, we deploy *DualMap* on top of vLLM (Kwon et al., 2023). All experiments are conducted on a distributed LLM serving cluster. Each node in the cluster is equipped with 8 Ascend NPUs (910B4: 32 GB HBM or 910B3: 64 GB HBM), and 1.5 TB DRAM.

**Metrics.** We evaluate the following: *Effective Request Capacity* (Qin et al., 2025), the percentage of requests with TTFT below a 5s SLO, reflecting system ability to handle latency-sensitive requests (Table 1); *Goodput*, the peak request rate sustained under required SLO (e.g., 90%), correlating with lower per-request cost (Zhong et al., 2024); *P50* and *P90 TTFT*, measuring overall scheduling/inference efficiency and tail latency in the prefill stage; *End-to-End Latency (E2E)*, total time from request arrival to final token generation, reported as P50 and P90; and *Cache Hit Rate* and *Load Balance Ratio*, assessing scheduling efficiency (§A.3.2). Elasticity evaluation is in §A.3.3.

**Baselines and Workloads.** We compare DualMap with four scheduling strategies: *Cache Affinity*, *Least Loaded*, *Min TTFT* (Qin et al., 2025), and *Preble* (Srivatsa et al., 2024). We use two real-world trace datasets from Mooncake (Qin et al., 2025): *Conversation* and *Tool&Agent*, detailed in §A.3.1.

**Model Deployment.** We evaluate Qwen2.5 7B and 14B models (Team, 2024) using default `float16` precision. Each instance is assigned a dedicated NPU (910B4 for 7B, 910B3 for 14B). Context caching is configured to store up to 1 million tokens for the 7B model and 0.5 million tokens for the 14B model, covering 30% and 15% of total request tokens, respectively.

### 4.2 END-TO-END PERFORMANCE

We evaluate the end-to-end performance of various scheduling strategies under two real-world workloads, using the Qwen2.5 7B and 14B models with 8 instances on different request rates (QPS).

**Effective Request Capacity and Goodput.** As shown in Figures 3b and 3d, on the *Tool&Agent* dataset, the presence of a large number of skewed prefixes makes the *Cache Affinity* strategy perform poorly. Even under low QPS, its SLO attainment rate is significantly lower than that of other strategies due to severe load imbalance, which leads to excessive queuing delays and frequent violations of the TTFT SLO. *Preble* and *Min TTFT* achieve suboptimal performance: their load-balancing capabilities mitigate the cluster imbalance caused by skewed loads, but at the cost of reduced cache hit rates. In contrast, *DualMap*, which simultaneously achieves cache affinity and load balancing,

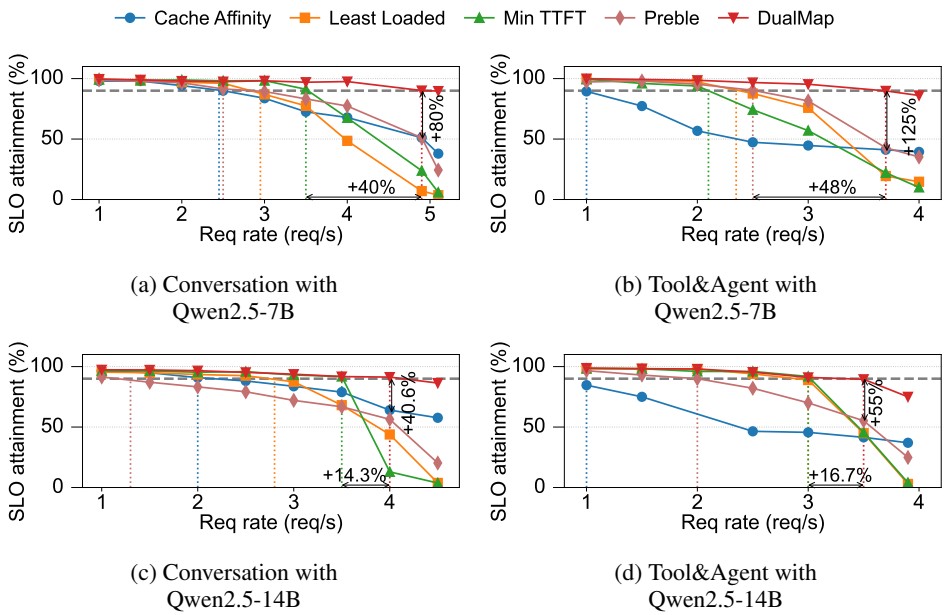

Figure 3: Effective request capacity and goodput of different scheduling strategies.

consistently outperforms other approaches. Its Effective Request Capacity increases by up to 125% (Figure 3b), while goodput improves by 16.7%-48% compared to the best baseline.

As shown in Figures 3a and 3c, on the *conversation* dataset, without skewed prefixes, *DualMap* still demonstrates significant advantages over other strategies. Its Effective Request Capacity increases by 40.6%–80%, and its goodput improves by 14.3%–40% compared to the best baseline.

**TTFT and E2E Latency.** As shown in Figure 4, across all experimental settings, *DualMap* significantly outperforms all baselines in both TTFT and E2E latency. Under high QPS scenarios, compared to the best baseline, *DualMap* reduces P50 TTFT by 55.4%–97.4% by achieving cache affinity, which yields high cache hit rates, effectively reduces redundant computation, and lowers TTFT; and reduces P90 TTFT by 82.3%–97% by achieving load balancing, which avoids request accumulation on individual instances and effectively reduces tail queuing time. The E2E latency closely follows the TTFT trends at both P50 and P90, as optimizations in the prefill stage indirectly impact the overall E2E latency. Comparing different models, *DualMap* consistently achieves lower TTFT and E2E latency than baselines on both the 7B and 14B models. Across varying QPS, *DualMap* maintains latency close to low-QPS levels even when QPS doubles, demonstrating strong stability, whereas baseline latencies increase rapidly with higher QPS.

## 4.3 ABLATION STUDY

To understand the source of DualMap's performance gains, we conduct an ablation study by incrementally enabling the techniques described in §3 under the Conversation workload using the Qwen2.5-14B model. We compare five configurations: *1) DualMap-cache-affinity*, uses cache-affinity between two candidates during DualMap initial routing (§3.2). *2) DualMap-least-loaded*, uses least-loaded selection; *3) DualMap-min-ttft*, uses min-TTFT selection. *4) DualMap-no-rebalance*, includes SLO-aware request routing but disables the hotspot-aware rebalancing; *5) DualMap*, includes both techniques introduced in §3.2 and §3.3.

As shown in Figure 5, *DualMap-cache-affinity* shows the highest P50 (Figure 5a) and P90 (Figure 5b) TTFT, as maximizing cache reuse (Figure 5c) causes severe load imbalance (Figure 5d) and long queuing delays. *DualMap-least-loaded* alleviates imbalance but suffers from low cache reuse, while *DualMap-min-ttft* slightly improves yet still has low cache hit rate due to frequent switching between cache-aware and load-aware scheduling. Compared with *DualMap-min-ttft*, *DualMap-no-rebalance* performs better, reducing P50 and P90 TTFT by 23.5% and 18.5% through SLO-aware

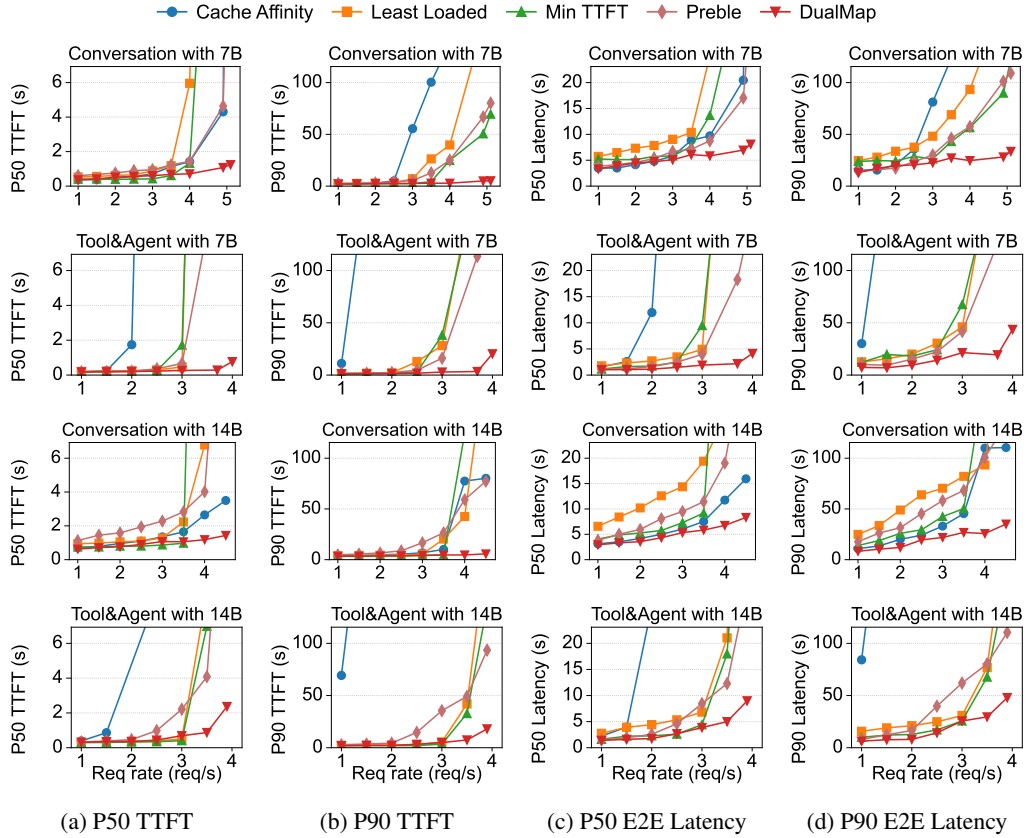

Figure 4: TTFT and E2E Latency of different scheduling strategies.

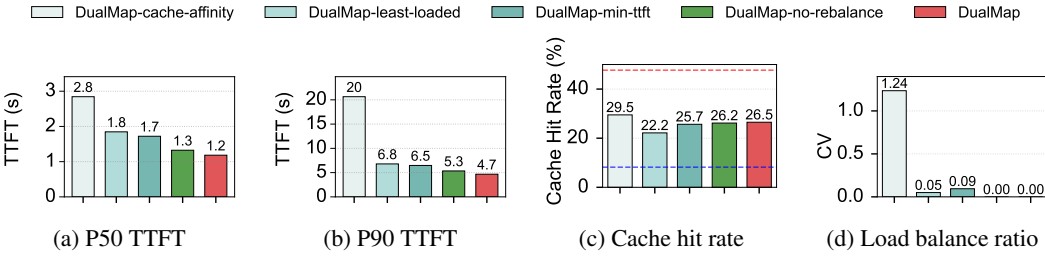

Figure 5: Ablation Results under the Conversation workload using the Qwen2.5-14B model.

scheduling. Finally, *DualMap* achieves the lowest latency, further reducing P90 TTFT by 11.3% with hotspot-aware rebalancing.

# 5 CONCLUSION

This paper addresses the conflict between cache affinity and load balancing in distributed LLM serving. We propose *DualMap*, a dual-mapping inference scheduler that simultaneously ensures KV cache reuse and balanced workload distribution. Key techniques include SLO-aware request routing, hotspot-aware rebalancing and lightweight dual-hash-ring scaling. Experimental results show that *DualMap* boosts effective request capacity by up to 2.25× under the same TTFT SLO constraints, and significantly reduces P90 latency across real-world workloads, compared with the state-of-the-art work.

## 6 ACKNOWLEDGEMENTS

This work was supported by National Key R&D Program of China NO.2023YFB4502801 and Huawei. We sincerely thank the anonymous reviewers for their constructive comments and valuable suggestions.

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

## A   APPENDIX

### A.1   LLM USAGE STATEMENT

In preparing this paper, we used a Large Language Model in polishing the writing. Specifically, the LLM was employed to improve grammar and readability of the text. No part of the research ideation, experimental design, data analysis, or results interpretation involved the use of LLMs. The authors take full responsibility for all content of the paper.

### A.2   DESIGN

#### A.2.1   SLO-AWARE REQUEST ROUTING

To balance cache reuse and load distribution, the *DualMap* global scheduler maps each request to two candidate instances using two independent hash functions over the request's prompt prefix.

Selecting between the two candidates involves a critical trade-off: prioritizing cache affinity reduces the prefill compute time through KV cache reuse, while prioritizing load balancing reduces queuing delays. This choice directly impacts the request's TTFT.

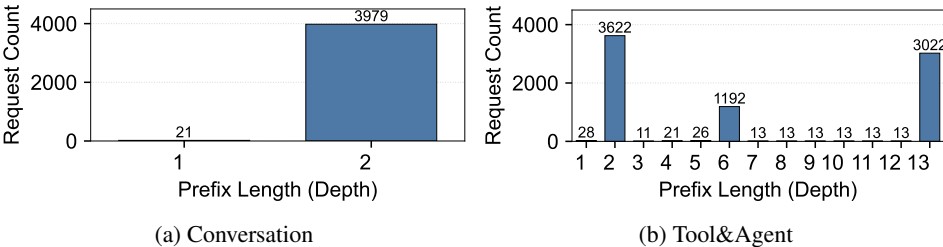

(a) Conversation

(b) Tool&Agent

Figure 6: Distribution of hash key prefix lengths for the *Conversation* and *Tool&Agent* datasets.

**How long prefix is appropriate for the hash key?** We analyze the distribution of hash key lengths on two datasets: *Conversation* and *Tool&Agent*, where the length represents the number of blocks (one block contains 512 tokens) in the prefix.

As shown in Figure 6a, on the *Conversation* dataset, 95% of requests have a shared prefix length of 2 blocks. This is because the prefixes in this dataset are not skewed. Therefore, two blocks (one system block and one user input block) are sufficient to identify shared-prefix requests, allowing them to be scheduled to the same instance without being scattered across multiple instances due to overlong hash keys.

In contrast, on the *Tool&Agent* dataset(Figure 6b), 45.2% of requests are similar to those in the *Conversation* dataset, with a hash key length of 2 blocks, because these requests are relatively balanced. However, 14.9% and 37.8% of requests have hash key lengths of 6 and 13 blocks, respectively. This is due to two abnormally popular prefixes in the *Tool&Agent* dataset. Our adaptive hash prefix length mechanism detects these skewed prefixes and extends their hash key lengths, preventing them from being mapped to the same instance and avoiding severe load imbalance.

**Baseline Strategies.** A naive least-loaded strategy selects the instance with the lower load, disregarding cache availability. As shown in Figure 7, Request 1 is routed to Instance 2 because it has fewer queued requests, even though Instance 1 already holds the matching KV cache. This results in a cache miss and longer prefill compute time, which in turn increases GPU occupancy and delays subsequent requests (e.g., Requests 3, 5, and 7), elevating their TTFT and increasing the risk of violating SLOs.

In contrast, a pure cache-affinity strategy always routes a request to the instance with the highest expected cache reuse. While this minimizes recomputation, it can lead to severe load imbalance. For example, if all matching prefixes are cached on Instance 1 as shown in Figure 7, all corresponding requests will be routed there, leading to queue buildup and long queuing delays (e.g., Requests 5, 6, and 7), despite short computation times.

To minimize the risk of TTFT SLO violations, the *Min TTFT* policy estimates TTFT for each candidate instance and selects the one with the lower value. For a request $r$ and candidate instance $i$, we define:

- $T_q(r, i)$: estimated queuing delay of Request $r$ on Instance $i$, capturing system load.
- $T_c(r, i)$: expected computation time for Request $r$ on Instance $i$, depending on whether cache reuse occurs.

Thus, the estimated total TTFT is:

$$TTFT(r, i) = T_q(r, i) + T_c(r, i) \tag{7}$$

In the example, although Instance 1 has a longer queuing delay ($T_q(1, 1) > T_q(1, 2)$), it benefits from cache reuse ($T_c(1, 1) \ll T_c(1, 2)$), resulting in lower total TTFT:

$$TTFT(1, 1) = T_q(1, 1) + T_c(1, 1) < TTFT(1, 2) = T_q(1, 2) + T_c(1, 2) \tag{8}$$

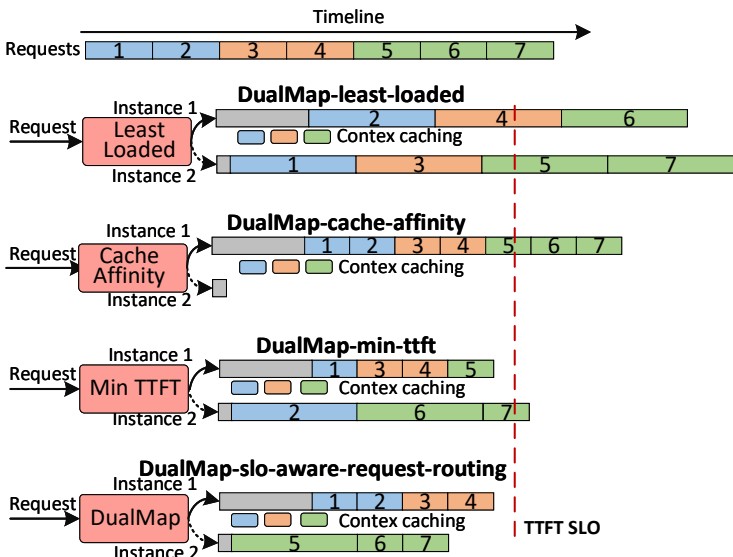

Figure 7: SLO-aware request routing. Different colors represent different prefixes. Instance 1 has context caching for all prefixes, while Instance 2 has no context caching.

Therefore, *Min TTFT* routes Request 1 to Instance 1.

However, as new requests arrive, this strategy may oscillate between cache-aware and load-aware decisions. For example, when Request 2 arrives, the queue at Instance 1 grows $T_q(2,1) \gg T_q(2,2)$, causing the scheduler to send the request to Instance 2 despite a cache miss. This load rebalancing improves fairness at the cost of recomputation. As the queue at Instance 1 shortens, future requests (e.g., Requests 3, 4, and 5) are again routed there, until the load imbalance grows again. In cases like Request 6, the scheduler again sacrifices cache affinity for load balancing.

While adaptive, this switching behavior leads to frequent cache misses under load fluctuation, increasing recomputation overhead and worsening overall TTFT. As observed with Request 7, repeated recomputation and queuing can lead to SLO violations. Fundamentally, Min TTFT's pursuit of optimal latency per request inadvertently degrades cache reuse and destabilizes load balance over time.

**The SLO-Aware Strategy.** To address these issues, *DualMap* adopts a TTFT-SLO-aware routing strategy that explicitly incorporates TTFT constraints. The key idea is to maintain cache affinity whenever possible, and only trade it off when load imbalance threatens to breach SLO constraints.

*DualMap* prioritizes cache affinity and initially selects the instance with the highest cache reuse. Requests are routed to this instance until its load causes the expected TTFT to exceed the SLO. At that point, *DualMap* switches to a load-aware strategy and selects the less loaded instance to avoid long queues and reduce TTFT tail latency. As shown in Figure 7, Requests 1–4 are routed to Instance 1 due to acceptable load difference and high cache reuse. Once the cache-affine instance becomes overloaded and violates the TTFT SLO, Request 5 is routed to Instance 2 to rebalance the load, even at the cost of recomputation.

Unlike *Min TTFT*, DualMap does not seek per-request optimal TTFT. Instead, it preserves cache reuse whenever load conditions permit, switching to load balancing only when necessary. This reduces the frequency of recomputation and stabilizes cache hit rates. Furthermore, if both instances have a matching prefix in their caches (i.e., equal prefix hit rate), *DualMap* always chooses the less-loaded one (e.g., Requests 6 and 7), further enhancing load balance without sacrificing reuse.

### A.2.2 HOTSPOT-AWARE REQUEST REBALANCING

**Instance-hotspots in system.** While the PoTC principle provides strong load-balancing guarantees under uniformly random choices (Mitzenmacher, 2002), real-world request patterns are often highly

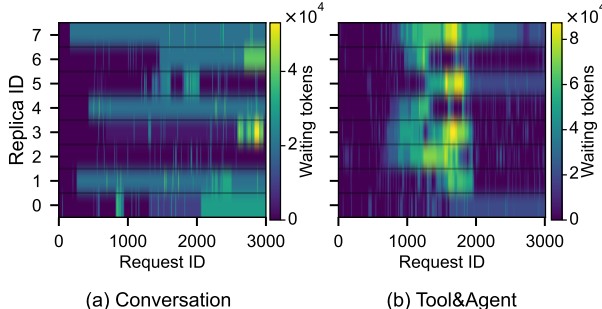

Figure 8: Hot instance phenomenon in real-world datasets. The bright yellow regions indicate overloaded instances.

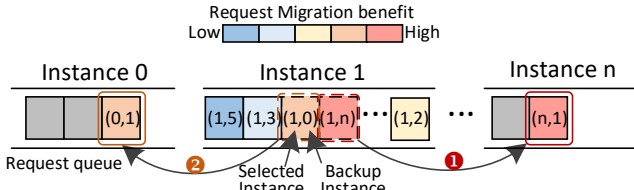

Figure 9: Hotspot-aware request migration.

skewed. In practice, KV prefix popularity often follows a skewed distribution (Wang et al., 2025b), making the candidate selection in *DualMap* non-uniform. For instance, tool-agent scenarios frequently involve a small number of widely used tools, while conversation agents often operate on trending topics. These skewed access patterns lead to certain prefixes being disproportionately common, causing some instances to be selected far more frequently than others, even under *DualMap*. As a result, persistent hotspots emerge. Empirical results in Figure 8 show that hot instances frequently emerge under *DualMap* scheduling, particularly in the tool-agent workload. This is attributed to the high concentration of requests targeting popular tool prompts, which are repeatedly hashed to the same candidate instances, resulting in severe queuing and load imbalance.

**Instance-hotspots-aware Rebalancing.** To achieve timely rebalancing, *DualMap* initiates batch migration during the initial routing phase whenever both candidate instances of a request are identified as overloaded. As illustrated in Figure 9, for each overloaded instance, *DualMap* estimates the migration benefit of each queued request using Eq. 6. A request is eligible for migration only if it satisfies satisfying $B_r^{(i \to j)} > 0$ and $\text{TTFT}_{r,j} < \text{TTFT}_{\text{SLO}}$, ensuring both TTFT improvement and SLO compliance. Requests are prioritized by descending benefit and migrated until all requests in the overloaded instance's queue are expected to meet the TTFT SLO. This hotspot-aware migration mechanism effectively relieves overloaded instances while preserving cache affinity.

### A.2.3  LIGHTWEIGHT DUAL-HASH-RING SCALING

In *DualMap*, all instances and request prefixes are placed on a logical ring based on their hash values, and each hashed prefix selects the nearest clockwise instance as its candidate. Thus, mappings depend solely on their relative positions on the ring.

Adding an instance introduces a new anchor point on the ring. Only requests whose hashed positions fall between the previous and the new anchor are remapped; all others retain their original mappings. Similarly, removing an instance affects only the prefixes that previously mapped to that instance.

We illustrate this with a simple example. Suppose the cluster has four instances $A, B, C, D$ placed on a logical ring according to their hash values, with the clockwise order $A \to B \to C \to D$. The resulting prefix-to-instance mapping is:

$$(A, B] \to B, \quad (B, C] \to C, \quad (C, D] \to D.$$

Requests hashed in $(A, B]$ map to $B$, those in $(B, C]$ to $C$, and those in $(C, D]$ to $D$.

Now suppose a new instance $E$ is added, and its hash value places it between $B$ and $C$, yielding the updated ring $A \rightarrow B \rightarrow E \rightarrow C \rightarrow D$. The mapping becomes:

$$(A, B] \rightarrow B, \quad (B, E] \rightarrow E, \quad (E, C] \rightarrow C, \quad (C, D] \rightarrow D.$$

Only requests whose hashed positions fall in $(B, E]$ are remapped to the new instance $E$, while all other mappings remain unchanged.

### A.3 EVALUATION

#### A.3.1 WORKLOADS

We use two real-world trace datasets from Mooncake (Qin et al., 2025): *Conversation* and *Tool&Agent*. *Conversation* is derived from multi-turn chatbot interactions. It exhibits approximately 40% prefix caching ratio due to high intra-session prefix reuse. *Tool&Agent* is characterized by long and repetitive system prompts, with a prefix cache ratio of 59%. Table 1 summarizes the workload characteristics. To ensure meaningful cache behavior under limited KV cache capacity, we evaluate the first 4,000 requests from *Conversation* and the first 8,000 from *Tool&Agent*. Arrival timestamps are preserved and scaled to simulate varying QPS levels.

Due to memory constraints on NPUs, the input length of each request is capped at 20,480 tokens for the 7B model and 10,240 tokens for the 14B model. To ensure consistent cache behavior, the first 500 requests are used for warm-up and excluded from all reported results.

Table 1: Workload characteristics for Conversation and Tool&Agent traces.

|  | Conversation | Tool&Agent |
|---|---|---|
| Avg. Input Length | 12,035 | 8,596 |
| Avg. Output Length | 343 | 182 |
| Prefix Caching Ratio | 40% | 59% |
| Number Requests | 4,000 | 8,000 |

#### A.3.2 CACHE HIT RATE AND LOAD BALANCE RATIO

To understand the source of performance differences across scheduling strategies, we evaluate the cache hit rate and load balance ratio of four baselines and *DualMap* under the *Conversation* and *Tool&Agent* workloads using the Qwen2.5-7B model.

**Cache Hit Rate.** As shown in Figures 10a and 10b, *DualMap* achieves cache hit rates comparable to the *Cache Affinity* strategy under both the *Conversation* and *Tool&Agent* workloads, reaching 62.5% and 96.4% of the theoretical upper bound, respectively. This is attributed to *DualMap*'s prompt-based mapping strategy and its cache-aware routing when conditions permit(§3.2). These designs enable *DualMap* to preserve strong cache affinity, thereby reducing prefill computation overhead, shortening the NPU occupation time per request, and decreasing queueing delays for subsequent requests—ultimately improving the likelihood of meeting TTFT SLOs.

In contrast, the *Least Loaded* strategy exhibits significantly lower cache hit rates on both workloads, as it prioritizes load balancing across instances without regard for cache reuse. *Min TTFT* improves upon this by incorporating cache-aware routing for a subset of requests, resulting in higher cache hit rates. *Preble* performs similarly to *Least Loaded* due to its conservative policy: it only engages in cache-aware scheduling when the request's prefix cache hit rate exceeds 50%, which limits its overall cache reuse potential.

**Load Balance Ratio.** As shown in Figures 11a and 11c, *DualMap* maintains consistently low and stable system load throughout the entire experiment, significantly outperforming all baselines. This benefit comes from its dual-mapping design, which achieves high cache affinity and consequently reduces overall computational load, enabling faster request completion. In addition, upon detecting overload, *DualMap* proactively migrates requests from overloaded instances to lighter ones. This allows *DualMap* to maintain load balance across all instances, ensuring efficient resource utilization and reduced queuing delays, as shown in Figures 11b and 11d.

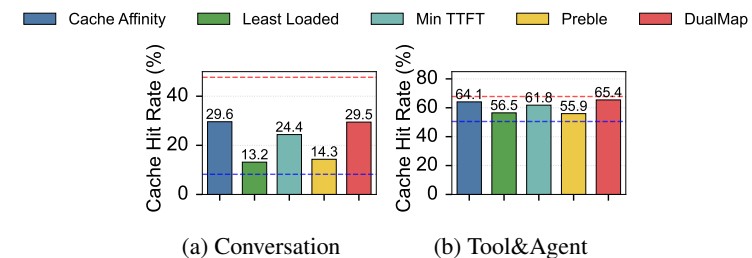

Figure 10: Cache hit rate with Qwen2.5-7B.

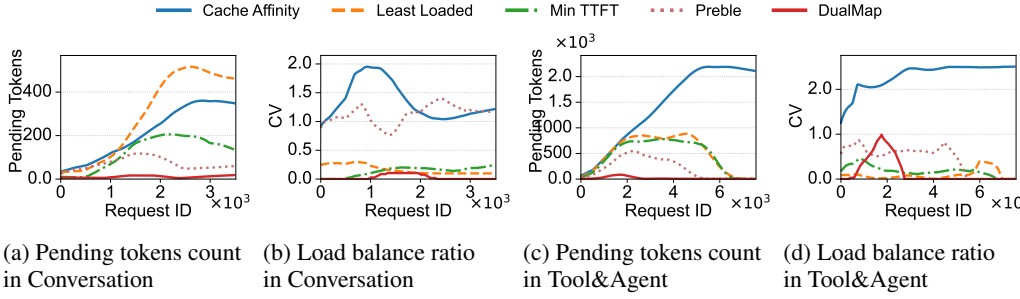

(a) Pending tokens count in Conversation    (b) Load balance ratio in Conversation    (c) Pending tokens count in Tool&Agent    (d) Load balance ratio in Tool&Agent

Figure 11: Pending tokens count and load balance ratio across all instances with Qwen2.5-7B.

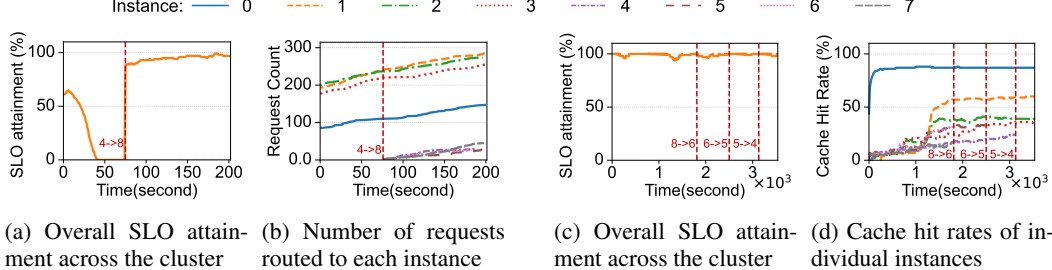

(a) Overall SLO attainment across the cluster    (b) Number of requests routed to each instance    (c) Overall SLO attainment across the cluster    (d) Cache hit rates of individual instances

Figure 12: Scaling experiments: (a) and (b) scale-up from 4 to 8 instances; (c) and (d) scale-down from 8 to 4 instances. Stability at 4 instances.

The load across instances is not always perfectly balanced. This is because *DualMap* prioritizes cache-aware scheduling among two candidate instances during initial routing, until the number of queued requests on cache-hitting instances grows too large to meet the TTFT SLO. This approach ensures that most requests still meet their TTFT SLO while maximizing cache reuse and reducing prefill computation. As a result, the number of pending tokens per instance is significantly reduced for *DualMap*. In contrast, *Cache Affinity* exhibits poor load balance, as its design co-locates requests with shared prefixes onto the same instance, resulting in severe skew, which leads to continuous accumulation of pending tokens. *Least Loaded* achieves an almost perfectly uniform load distribution (CV=0) on both workloads, consistent with its load-aware scheduling policy. However, its pending token count is much higher than that of *DualMap* due to insufficient cache reuse. *Min TTFT* and *Preble* show noticeable improvements over pending tokens count and CV by trading off load balancing and cache affinity.

### A.3.3 ELASTICITY EVALUATION

We evaluate *DualMap*'s elasticity under the *Tool&Agent* workload using the Qwen2.5-7B model. *SLO attainment* (i.e., percentage of requests with TTFT < 5s) is used to assess scaling effectiveness.

**Scaling Up.** As shown in Figure 12a, with 4 instances under QPS=4, the system quickly becomes overloaded, causing widespread SLO violations. At 74 seconds, *DualMap* detects overload and

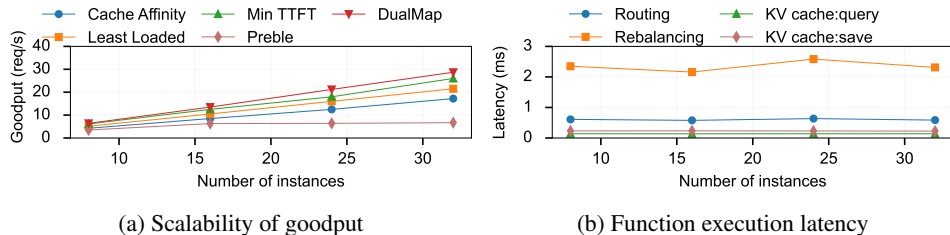

(a) Scalability of goodput          (b) Function execution latency

Figure 13: Scalability and scheduler overhead under the *Tool&Agent* workload using the Qwen2.5-7B model, with the number of instances varying from 8 to 32.

adds 4 instances (Instance 4–7). SLO attainment promptly rises to 90% as the scheduler routes new requests to the newly added idle instances (Figure 12b), relieving pressure on hot instances.

**Scaling Down.** Later, under reduced load (QPS=2) and with 8 instances, the system becomes underutilized. At 1809 seconds, *DualMap* begins gradual downscaling, reducing to 4 instances (Figure 12c), while maintaining over 90% SLO attainment.

### A.4 Scalability and Overhead

We evaluate the scalability of *DualMap* and scheduler overhead using the Vidur-based simulator (Agrawal et al., 2024). We simulate a cluster of instances, each with 64 GB DRAM and one NPU (910B4), serving Qwen2.5-7B on the *Tool&Agent* workload. The number of instances is scaled from 8 to 32, and the total number of requests is increased proportionally from 8K to 32K.

#### A.4.1 Scalability Analysis.

As shown in Figure 13a, *DualMap* achieves near-linear growth in goodput (the maximum sustainable request rate under the 90% TTFT SLO) (Zhong et al., 2024) and consistently outperforms Cache Affinity, Least Loaded, Min TTFT, and Preble across all cluster sizes. This demonstrates the strong scalability of the dual-mapping design. These results indicate that DualMap's dual-mapping design scales well: prefix-bound dual hashing preserves cache affinity, while rebalancing always operates within candidate pairs, keeping the load-balancing behavior stable as the cluster grows.

#### A.4.2 Scheduler Overhead.

*DualMap* maintains per-instance metadata for DRAM-based KV caches and request queues, updating it as requests are scheduled and executed. KV cache metadata supports estimating reusable tokens and thus prefill computation cost, while queue metadata provides queueing-delay estimates for TTFT prediction, enabling both SLO-aware routing and hotspot-aware rebalancing. The primary overhead of *DualMap* arises from the metadata footprint and the runtime footprint of key scheduler operations.

**Metadata footprint.** Metadata footprint is dominated by KV cache (request queues are typically short, so their cost is negligible). Following MooncakeAPC-style (202, 2025) designs, each 128-token block stores a hash value and block ID (8B each), i.e., 16B per block. For Qwen2.5-7B, a 64GB KV cache contains 9,632 blocks, so each instance stores about 146.2 KB of metadata, which is engineering-wise negligible; a 32-instance cluster stores 4.57MB. For Qwen2.5-14B, the per-instance and 32-instance metadata costs are 44KB and 1.38MB, respectively, growing linearly with total KV-cache capacity.

**Runtime footprint.** *DualMap's* runtime footprint primarily comes from three operations (Figure 13b): KV cache access, SLO-aware request routing(§3.2), and hotspot-aware request rebalancing(§3.3).

(1) *KV cache access.* For each request, *DualMap* computes the number of reusable tokens by querying the KV cache to estimate the remaining prefill computation and expected TTFT. Both KV cache query and save operations take $\sim$0.2 ms and are independent of cluster size, since each instance's KV cache metadata is maintained and accessed independently by the global scheduler.

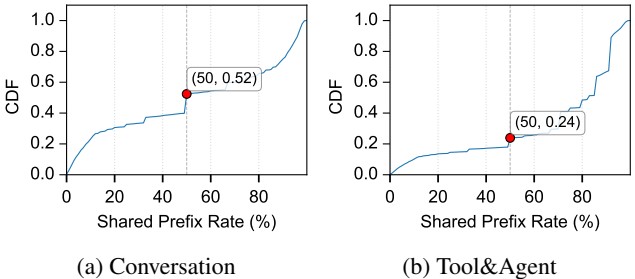

(a) Conversation    (b) Tool&Agent

Figure 14: CDF of shared prefix rate for the *Conversation* and *Tool&Agent* workload.

(2) *Routing(§3.2)*. SLO-aware request routing takes ∼0.6 ms per request, consisting of dual hashing to obtain candidate instances (∼0.05 ms) and TTFT estimation via KV cache queries. These operations depend only on per-instance metadata, so routing latency does not grow with cluster size.

(3) *Rebalancing(§3.3)*. Hotspot-aware request rebalancing takes 2.2–2.5 ms per invocation. For each request on an overloaded instance, *DualMap* computes the current TTFT and the hypothetical TTFT after migration. Each TTFT evaluation requires two KV cache queries (one on the overloaded instance and one on its backup), so the total cost scales with the queue length of the overloaded instance, independent of the total number of instances in the cluster. In practice, instance request queues are typically short, so the overhead of hotspot-aware rebalancing remains bounded and acceptable.

## A.5   SHARED PREFIX RATE CHARACTERIZATION

Prompts are partially shared across requests in real-world workloads (Srivatsa et al., 2024; Gao et al., 2024). For the workloads used in our study, *Conversation* and *Tool&Agent*, we measure the *shared prefix rate*, defined as the fraction of a request's prompt that overlaps with the longest shared prefix among preceding requests within the same workload, where "length" refers to the number of tokens. Figure 14a shows that 48% of *Conversation* requests share at least 50% of their prompt prefix, reflecting naturally recurring dialogue histories across multiple conversation turns. Figure 14b shows an even stronger effect: 76% of *Tool&Agent* requests share at least 50% of their prefix, largely due to repeated system prompts and consistent tool prompts. These results confirm the high prevalence of shared prefixes in practical LLM serving environments and highlight the importance of cache-affinity-aware scheduling to maximize KV cache reuse and reduce prefill computation.

## A.6   INTEGRATION INTO DISAGGREGATED ARCHITECTURES

*DualMap* is agnostic to the internal micro-architectural design of the serving system and only requires an estimated TTFT for each candidate instance.

**Prefill–Decode (PD) Disaggregation.** In PD-disaggregated systems, TTFT is dominated by the prefill phase. *DualMap* therefore performs routing exclusively on the prefill instances, while decode instances are attached following the serving framework's existing mechanism (e.g., vLLM). This design avoids any invasive changes to the underlying execution pipeline.

**Attention–FFN (AF) Disaggregation.** AF disaggregation is primarily an optimization for the decode stage (Zhu et al., 2025; Wang et al., 2025a). In practice, many systems first apply PD disaggregation and then perform AF disaggregation within the decode nodes. Similar to the PD setting, *DualMap* focuses on scheduling the prefill (P) instances, where TTFT is determined by prefill computation and queueing delay. Because AF disaggregation occurs downstream in the decode stage, it does not affect *DualMap*'s core routing objective: minimizing TTFT through KV cache affinity and load balancing on the prefill instances. Thus, *DualMap* naturally remains compatible with AF-disaggregated architectures without requiring additional modifications.

### A.7 CLARIFYING NOVELTY AND CONTRIBUTION

While *DualMap* adopts hashing primitives similar to classical distributed systems such as Dynamo (DeCandia et al., 2007) and Chord (Stoica et al., 2001), it addresses a fundamentally different challenge specific to distributed LLM serving: **reconciling KV cache affinity with load balancing** under strict TTFT SLOs. While both *DualMap* and these systems use hashing primitives, their mechanisms diverge significantly:

- **Novel Mechanism: Prefix-Bound Dual Candidates.** Classical systems map each key to a single location (e.g., Chord's successor) or a primary with fixed replicas (e.g., Dynamo). *DualMap* introduces a *prefix-bound dual-candidate* mechanism:

$$\text{Prefix}(p) \Rightarrow \{I_1, I_2\}.$$

  Each request prefix $p$ is deterministically mapped to a *pair* of candidate instances via two independent hash functions. This provides exactly two degrees of freedom for load balancing—following the PoTC principle—while ensuring that all requests sharing the same prefix are always routed among the same two candidates. This structure is essential for maintaining KV-cache locality and is not present in traditional systems.

- **SLO-Aware Request Routing.** Instead of static replica selection, *DualMap* chooses between its two candidates using a TTFT estimation function that jointly considers the *cache-reuse benefit* and *queuing delay*, which is entirely absent in traditional systems.

- **Hotspot-Aware Request Rebalancing.** LLM workloads often display highly skewed request distributions, resulting in concentrated instance hotspots that may violate SLOs by consistent hashing alone. *DualMap* introduces *hotspot-aware request rebalancing*, which selectively migrates requests *only within their prefix-bound candidate pair*. This migration alleviates overload without scattering prefixes across instances, preserving cache locality—a property that traditional systems neither require nor are designed to support.

*DualMap* is not a minor variant of consistent hashing. It is a novel scheduling framework purpose-built to simultaneously achieve cache affinity and load balancing under the strict SLO requirements of modern LLM serving workloads.

### A.8 DISCUSSION: ROBUSTNESS OF TTFT ESTIMATION UNDER MEMORY CONTENTION

This section provides additional details on the robustness of TTFT estimation in scenarios with NPU memory contention.

#### A.8.1 MEMORY-EXHAUSTION-INDUCED DECODE BOTTLENECK

Under the vLLM serving backend, each inference instance prioritizes prefill computations over decodes as long as sufficient NPU memory is available. Memory is released only after a request completes its decode stage. At high request rates, repeated prefills cause "rapid memory consumption on the NPU, eventually preventing new prefills from being scheduled. Once this occurs, the scheduler must switch to decode execution in order to free memory blocks. During this phase, a prefill request may need to wait for one or more decode tasks to finish, causing a delay whose magnitude depends on the lengths of the ongoing decodes as well as the current memory pressure. We refer to this phenomenon as the memory-exhaustion-induced decode bottleneck. The delay introduced in this state is denoted as $D_i$ for instance $i$.

#### A.8.2 IMPACT OF THE DECODE BOTTLENECK ON SLO-AWARE REQUEST ROUTING

SLO-aware request routing uses the original TTFT estimate $TTFT(r, i) = T_q(r, i) + T_c(r, i)$, where $TTFT(r, i)$ denotes the predicted TTFT for dispatching request $r$ to instance $i$. When the decode bottleneck state introduces an additional delay $D_i$, the actual TTFT becomes $TTFT_{r,i}^{\text{actual}} = D_i + T_q(r, i) + T_c(r, i)$. If $D_i$ is small, $TTFT_{r,i}^{\text{actual}}$ typically remains below the TTFT-SLO threshold, causing negligible deviation in SLO attainment. In contrast, when $D_i$ becomes large, $TTFT_{r,i}^{\text{actual}}$ is likely to exceed the TTFT-SLO, causing most requests on the memory-exhausted instance to violate TTFT SLO. This behavior effectively mirrors an overloaded-instance condition, in which

requests fail to meet their TTFT-SLO. Consequently, we delegate the mitigation of persistent decode bottleneck states to the hotspot-aware request rebalancing mechanism.

### A.8.3 HOTSPOT-AWARE REQUEST REBALANCING UNDER DECODE BOTTLENECKS

In the hotspot-aware request rebalancing mechanism, an instance that remains in a decode-bound phase for an extended period is treated as overloaded, and requests in its queue that are at risk of violating the TTFT SLO should be migrated to less-loaded instances, thereby mitigating the issue of widespread TTFT SLO violations caused by decode bottlenecks. This mechanism relies on two core components: bottleneck detection and TTFT correction.

**Decode Bottleneck Detection.** An instance is considered to be in a decode bottleneck state when it fails to complete any prefill computation for an extended period while its request queue remains non-empty. To detect this condition, the global scheduler monitors the interval *prefill_interval* = $t_{\text{current}} - t_{\text{last\_prefill}}$, where $t_{\text{last\_prefill}}$ is the timestamp of the last prefill completion. Once this interval exceeds a threshold $T$, the instance is flagged as being in the long decode bottleneck state. This heuristic provides a practical approximation of the decode-induced delay and does not depend on the decode lengths of individual requests.

The threshold $T$ reflects a trade-off between timely detection of decode bottlenecks and the preservation of cache affinity. A large $T$ delays bottleneck detection, potentially increasing the number of SLO violations, whereas a small $T$ leads to frequent migrations that may undermine the cache affinity established by SLO-aware request routing. Based on empirical observations in our deployment, $T$ is set to 3 seconds, which works well in practice.

**Corrected TTFT Estimation and Request Migration.** For an instance identified as a decode bottleneck hotspot, the TTFT estimate for its queued requests is corrected by adding an approximation of the decode delay: $TTFT_{r,i}^{\text{corrected}} = D_{\text{estimated}} + T_q(r,i) + T_c(r,i)$. Since the true $D_i$ cannot be determined accurately, *DualMap* approximates $D_{\text{estimated}}$ using the observed *prefill_interval*. Although this approximation may not be entirely accurate, the interval grows as the bottleneck state persists, enabling *DualMap* to identify requests that are at risk of exceeding the TTFT SLO. When the corrected TTFT of a request exceeds the SLO, the rebalancing mechanism migrates it to a less-loaded candidate instance, ensuring that the request meets the TTFT SLO. This mechanism effectively converts unpredictable decode bottleneck latency into an overload signal that the global scheduler can detect through the responses of individual instances executing requests.

Although the decode bottleneck state complicates precise TTFT estimation, the rebalancing mechanism ensures robustness by treating such states as overload conditions and migrating requests accordingly. This design enables *DualMap* to maintain high SLO attainment even under NPU memory contention that introduces decode-latency variability.

### A.9 OPTIMAL CANDIDATE SET SIZE

To determine the optimal candidate set size $d$, we compute the maximum load deviation according to PoTC theory. The number of inference instances ranges from $n = 8$ to $n = 32$, and the total number of requests $m$ is set to 8000–32000 such that the average load $\frac{m}{n}$ remains constant. As shown in Figure 15, increasing $d$ from 1 to 2 produces a sharp reduction in maximum load deviation, consistent with the well-established near-exponential gain enabled by two-choice load balancing. When $d$ exceeds 2, the marginal benefit quickly tapers off, indicating that larger candidate sets contribute little additional improvement in reducing imbalance.

These trends persist across all evaluated cluster sizes. Overall, $d = 2$ provides an effective balance for *DualMap*: it delivers strong inter-instance load balancing while keeping the candidate set small, thereby preserving high KV cache reuse.

## B RELATED WORK

**Prefix Sharing between Requests.** A common scenario among multiple requests is the sharing of prompts. For instance, requests within the same session in multi-turn conversations share the prompt history of previous requests (Sha, 2025; Gao et al., 2024; Zheng et al., 2024b; Yang et al.,

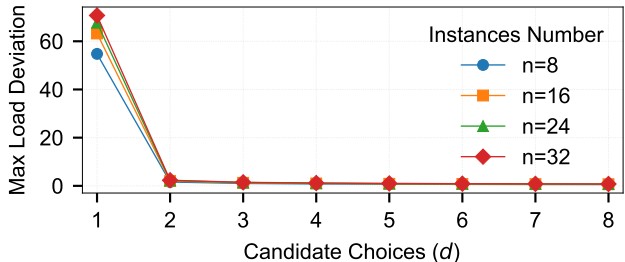

Figure 15: Maximum load deviation for different choices across cluster sizes ($n = 8$–$32$).

2025; Li et al., 2024a); in tool agents, requests accessing the same tool often have identical tool usage instructions (Hao et al., 2023). For these shared prompts, using KV cache (Kwon et al., 2023; Prabhu et al., 2025; Chen et al., 2025; Lee et al., 2024; Wu et al., 2024; Li et al., 2024b; Wu et al., 2025; Wang et al., 2025b; Sun et al., 2024; Strati et al., 2024; Wang et al., 2024; Ye et al., 2024; Yao et al., 2025; Agarwal et al., 2025; Shi et al., 2024; Zhang et al., 2025; Chen et al., 2024a;b) to trade computation for storage, by reusing the prefix's key-value (KV) tensors across multiple requests (202, 2025; Zheng et al., 2024a), allows for direct loading of the KV cache corresponding to these shared prompts without recomputation. This saves valuable GPU computing resources and significantly reduces inference latency.

**Scheduling Work Balancing *Cache Affinity* and *Load Balance*.** *Preble* (Srivatsa et al., 2024) adopts a heuristic strategy: When the prefix hit rate of a request (i.e., the ratio of the number of tokens matched with shared prefixes in the system to the total number of input tokens) exceeds 50%, the request is dispatched to the instance with the highest prefix hit rate, thus favoring *Cache Affinity*. Conversely, when the prefix hit rate is low, *Preble* routes requests based on a combination of request inference cost and current instance load. However, this strategy can lead to behavior similar to the *Least Loaded* policy. For example, consider two instances: *Instance 1* contains the KV cache for *Request 1*, and *Instance 2* contains the KV cache for *Request 2*. If *Instance 1* is more loaded when *Request 1* arrives, it will be routed to *Instance 2*, breaking cache locality. Later, *Request 2* may be routed to *Instance 1* for similar reasons.

*Dynamo* (Dyn, 2025) formulates a scheduling cost function to balance cache reuse against load balancing, defined as $\max_i(\text{KVMatch}_i - \text{Load}_i)$. Here, $\text{KVMatch}_i$ denotes the prefix hit rate of request $r$ on instance $i$, and $\text{Load}_i$ represents the current load on instance $i$. When the load difference between instances is large, the cost function is dominated by the load term, degenerating into a *Least Loaded* policy. Conversely, when the request's prefix hit rate is high, the KVMatch term dominates, resulting in behavior similar to *Cache Affinity*.

*Mooncake* (Qin et al., 2025) routes each request to the instance with the lowest estimated TTFT, which is composed of two factors: the queuing delay (proportional to the instance's current load) and the recomputation cost (inversely related to cache reuse). We refer to this policy as *Min TTFT* and simplify its cost function as $\max_i(\text{pending\_tokens\_count}_i - \text{recompute\_tokens\_count}_i)$. Where $\text{pending\_tokens\_count}_i$ denotes the number of tokens queued for prefill on instance $i$, and $\text{recompute\_tokens\_count}_i$ is the number of input tokens for request $r$ that must be recomputed on instance $i$ due to lack of KV cache reuse. Similar to *Dynamo*, when the system is heavily loaded, the cost is dominated by the queuing term $\text{pending\_tokens\_count}_i$, degenerating into the *Least Loaded* policy; when the system is lightly loaded, the cost is dominated by recomputation, behaving like *Cache Affinity*.

However, all these methods operate within a single mapping space and are fundamentally limited to balancing the trade-off between cache affinity and load balance. In contrast, *DualMap* proposes a novel dual-mapping scheduling framework enabling the system to simultaneously achieve both objectives.

