# OpenReview forum: "DualMap: Enabling Both Cache Affinity and Load Balancing for Distributed LLM Serving"
_ICLR.cc/2026/Conference — ICLR 2026 Poster_

### Official Review · Reviewer_J9qz · 2025-10-28

**Soundness:** 4
**Presentation:** 4
**Contribution:** 2
**Rating:** 6
**Confidence:** 5

**Summary:**

The paper addresses the challenge of LLM serving systems that utilize KV cache reuse for popular requests. The authors observe that existing systems often face a trade-off, prioritizing either nodes with high cache affinity or nodes that offer better load balancing. The paper argues that these two objectives can be jointly optimized, introducing a system called DualMap. The key insight is the application of the _power of two choices_ technique, where two nodes are probed for each request, and the request is assigned to the more suitable one. The suitability metric is designed to intelligently switch between prioritizing cache affinity or load balancing based on the current system state. Additionally, the authors propose active request rebalancing and the use of consistent hashing to facilitate scaling. The evaluation using two datasets shows that DualMap outperforms existing baselines.

**Strengths:**

- The paper is well-written, clear, and concise.
- It addresses an important and timely problem in LLM serving: the trade-off between cache affinity and load balancing.
- The proposed solution is systematic and effectively integrates established ideas from the systems literature.

**Weaknesses:**

- The core problem—hashing, server selection to optimize an objective, and scaling—has been studied extensively in the systems community. The proposed solution bears a strong resemblance to the work in [1]. This raises questions about the work's novelty. While _the application_ (LLM serving) is new, the underlying problem of load-aware request routing seems very similar. The paper would be significantly strengthened if the authors clearly delineated what differentiates DualMap from systems like [1], Chord~[2], or standard consistent hashing. What specific aspects of the LLM serving problem required novel design choices that existing solutions could not address? Explicitly positioning the work, even as a novel application and integration of established techniques to a new domain, would clarify its contribution.

- The assumption that the TTFT of a request $r$ can be accurately estimated a priori (using $TTFT_q$​ and $TTFT_c$​) seems overly simplistic. Request processing time is a complex function, not just of queue length or cache presence. Specifically, the execution time of a batch containing cache misses depends on the mix of prefill and decode requests within it, which is unknown at routing time. Therefore, estimating $T_q​(r, i)$ or $T_c​(r, i)$ just from the queue state seems insufficient. This estimation problem is compounded if the instance-level scheduler does not use a simple FCFS policy. Given these factors, how can the system reliably estimate TTFT and predict potential SLO violations at the moment of the routing decision?

Minor Comments:
- The paper's motivation rests on the importance of KV cache reuse. However, this claim is not substantiated with data. How frequently do large prefix overlaps occur in real-world workloads? Please add a citation or, preferably, a microbenchmark/workload-characterization figure (similar to those in related work) to demonstrate the performance impact and prevalence of this problem.
- Figure 1, which shows the trade-off, would be more effective as a Pareto curve rather than a bar chart. Please also plot DualMap's position on this curve to contextualize its design point.
- Equation 2 uses the term $L(s_i​)$, which is presumably the load of instance $i$. Please define this term explicitly.
- Figure 3 is central to the paper but is difficult to read due to cluttered text. Please consider replotting it, perhaps using a smaller font or simplifying the diagram.
- Figure 7 is key for understanding the SLO-aware strategy but is currently unclear. Please label the strategies directly on the figure to show how they map to the baselines and DualMap. This would make the figure self-contained, removing the need to cross-reference the text (e.g., line 729). Figure 7 in the Sarathi Serve paper is a good example of this.


[1] DeCandia, Giuseppe, Deniz Hastorun, Madan Jampani, Gunavardhan Kakulapati, Avinash Lakshman, Alex Pilchin, Swaminathan Sivasubramanian, Peter Vosshall, and Werner Vogels. "Dynamo: Amazon's highly available key-value store." ACM SIGOPS operating systems review 41, no. 6 (2007): 205-220.

[2] Stoica, Ion, Robert Morris, David Karger, M. Frans Kaashoek, and Hari Balakrishnan. "Chord: A scalable peer-to-peer lookup service for internet applications." ACM SIGCOMM computer communication review 31, no. 4 (2001): 149-160.

**Questions:**

1. What qualifies as a _sufficiently positive_ benefit $B_r(i \rightarrow j)$​ for triggering an active request rebalancing? Is there a specific threshold, and if so, how is it determined or tuned?
2. In the Appendix (line 665), it is stated that 95% of Conversation dataset requests share a prefix of 2 blocks. What is a block here? What is the size of a block in terms of tokens?

---

> ### Author Response · Authors · 2025-11-21
> **Official Comment by Authors**
>
> We are grateful for your careful and insightful review, especially your comments on novelty, TTFT estimation, and motivation for KV-cache reuse. We have revised the paper to clarify these aspects.
>
> ### **Weakness 1: Clarifying Novelty and Contribution**
>
> DualMap addresses a core problem unique to LLM serving: reconciling KV Cache affinity with load balancing under strict TTFT SLOs.
>
> While both DualMap and prior systems use hashing, their mechanisms differ:
>
> - **Novel Mechanism: Prefix-Bound Dual Candidates:** Traditional systems map a key to a single location (Chord) or primary plus fixed replicas (Dynamo). DualMap maps each prefix $p$ to two instances $(I_1, I_2)$ via independent hashes. This design is crucial because it provides the necessary degree of freedom (two choices) for load balancing, while simultaneously preserving the required cache affinity constraint (always selecting from the two predetermined candidates).
>
> - **SLO-aware Request Routing:** Unlike static replication, DualMap selects between the two candidates using a TTFT estimation function that trades off cache hit benefits against queuing costs to minimize estimated TTFT.
> - **Hotspot-Aware Rebalancing:**  LLM workloads often display highly skewed request distributions, resulting in concentrated instance hotspots that may violate SLOs by consistent hashing alone. DualMap tackles this challenge through Hotspot-Aware Rebalancing, which proactively migrates requests between the candidate pair to relieve severe hotspots—a capability that traditional systems neither require nor can feasibly support.
>
> We now emphasize in Appendix A.7 that DualMap is not a variant of consistent hashing but a framework designed to achieve cache affinity and load balancing in modern distributed LLM serving systems.
>
> ### **Weakness 2: Per-Request TTFT Estimation**
>
> We thank the reviewer for raising concerns about TTFT estimation. Our simple model focuses on the main bottlenecks: queuing delay and prefill recomputation cost.
>
> **Negligible Decode Impact:** The estimation ignores the decode stage, which is reasonable: under vLLM's prefill-priority scheduling, a prefill request waits at most for a single decode batch (milliseconds), negligible compared to the waiting time for other prefill requests, typically on the order of seconds.
>
> **FCFS policy:**   Although vLLM and SGLang use dynamic batching and non-strict FCFS, prefill requests follow an approximate FCFS principle, ensuring a strong correlation between waiting time and preceding prefill token workload, validating the queuing component in TTFT.  To remove uncertainties from backend schedulers (e.g., non-FCFS), DualMap will introduce a global request queue for each instance in future work. Both *SLO-aware Request Routing* and *Hotspot-Aware Rebalancing* operate on this global queue. When an instance's local queue is empty, the global scheduler dispatches a request from the global queue to the instance. This design separates DualMap's global FCFS scheduling from the backend scheduler 's execution optimizations, ensuring that the "number of tokens in the queue" used in TTFT remains a reliable indicator.
>
> While $\text{TTFT}$ is an approximation, DualMap's design ensures that it captures the dominant delay components accurately and remains robust against local scheduler variations, providing reliable predictions for potential SLO violations.
>
> ### **Minor Comments**
>
> We thank the reviewer for the valuable suggestions. All minor comments have been addressed in the revised manuscript:
>
> - **Prefix Sharing Motivation:** Added a subsection with CDF plots (Appendix A.5) quantifying shared-prefix rates.
> - **Pareto Curve:** Updated Figure 1.
> - **Definition of $L(s_i)$:** We have standardized the notation for an instance from $s_i$ to $I_i$ (Instance $i$), as the symbol $I$ is used to denote instances in other parts of the paper. Unified instance notation and explicitly defined $L(I_i)$ in Eq. (2).  $L(I_i)$ denotes the number of requests assigned to instance $I_i$
> - **Re-plotted Figure 3** for improved clarity.
> - **Revised Figure 7** with explicit in-figure labels for better readability.
>
> ### **Q1: Migration Benefit for a Request**
>
> During rebalancing, a request is migrated only if $\text{B}_{r}^{(i \rightarrow j)} > 0$, ensuring TTFT reduction. Requests are prioritized by benefit and migrated until the overloaded queue meets the TTFT SLO, with no manual threshold needed.
>
> ### **Q2: Clarification on the "Block" in the Appendix (line 665)**
>
> A block is a unit of 512 tokens for KV cache management (Mooncake[1]). Thus, 95% of requests sharing a prefix of 2 blocks means they share the first 1024 prefill tokens.
>
> We appreciate your time and remain available to answer any further questions.
>
> ### **References**
>
> [1] https://github.com/kvcache-ai/Mooncake/blob/main/FAST25-release/Mooncake-FAST25.pdf

---

> > ### Comment · Reviewer_J9qz · 2025-11-21
> >
> > Thank you for your response.
> >
> > Could you please verify that you have uploaded the latest manuscript? I can only see the old paper with the last updated date of 08 Oct 2025.

---

> > > ### Author Response · Authors · 2025-11-21
> > > **Official Comment by Authors**
> > >
> > > Thank you for your message. We apologize for the confusion. We have now uploaded the latest manuscript

---

> > > > ### Comment · Reviewer_J9qz · 2025-11-25
> > > >
> > > > I really appreciate the authors' efforts to address the comments from myself and the other reviewers; I believe the paper is now in much stronger shape. I also agree with your updated arguments regarding novelty and recognize that DualMap holds significant value for both the ML and systems communities.
> > > >
> > > > On TTFT Estimation: I remain somewhat conflicted on this point. While I agree that simplifying assumptions are necessary for estimation, I find them difficult to apply in real-world scenarios. For instance, you noted that vLLM prioritizes prefills over ongoing decodes, implying a new prefill request waits for at most one decoding cycle. However, this assumes sufficient memory is available to allocate blocks for the new request. If memory is contended, vLLM must either preempt an ongoing decode (potentially incurring overhead to move the KV cache to persistent storage) or wait for a decode to finish. Both scenarios introduce unpredictable latency, making precise TTFT estimation highly challenging.
> > > >
> > > > Nits:
> > > > - Figure 1: I suggest removing the dotted line, as the baselines are not connected to each other.
> > > > - Optimal d Analysis: I really enjoyed the new section explaining why two choices is the optimal point (as well as your response to the other reviewer). It would be great to add a plot showing the gain as you increase $d$ to visually demonstrate that $d=2$ is indeed the optimal configuration.

---

> > > > > ### Author Response · Authors · 2025-11-28
> > > > >
> > > > > We sincerely thank you for your continued engagement and for your positive feedback on the improved clarity and strength of our manuscript. We are extremely grateful for the insightful critique regarding the uncertainties in TTFT estimation, as well as the helpful suggestions concerning Figure 1 and the optimal $d$ analysis.
> > > > >
> > > > > ### **TTFT Estimation**
> > > > >
> > > > > We also empirically observed this exact phenomenon in our system implementation and high-load experiments: Under the vLLM framework, prefill is prioritized, and GPU memory is only released upon request decode completion. At high QPS, sustained prefill execution rapidly depletes GPU memory, forcing the scheduler to switch priority to decode in order to free up memory blocks. We refer to this as the **“memory-exhaustion-induced decode bottleneck.”** Although the presence of waiting time ($D$) in this state introduces inaccuracy into TTFT estimation, a small $D$ has a negligible impact on SLO attainment, whereas a large $D$ may cause requests on the memory-exhausted instance to violate the TTFT SLO. Such large-$D$ behavior effectively mirrors an overloaded-instance condition. Crucially,  our hotspot-aware request rebalancing mechanism (§3.3) was designed from the outset not only to rebalance  general overloaded instances but also to handle this decode bottleneck challenge.
> > > > >
> > > > >
> > > > >
> > > > > We acknowledge that the initial submission focused on the core novelty of DualMap in achieving both cache affinity and load balancing, and thus simplified the discussion of these considerations. To fully address your concern, we have added a detailed explanation in Appendix A.8 of the revised version.
> > > > >
> > > > > ### **Nits**
> > > > >
> > > > > We have updated Figure 1 based on your suggestion and added the plot for the optimal $d$ analysis to Appendix A.9.
> > > > >
> > > > >
> > > > >
> > > > > Thank you again for your thoughtful comments and for helping us strengthen the paper.

---

### Official Review · Reviewer_6DK7 · 2025-10-28

**Soundness:** 3
**Presentation:** 3
**Contribution:** 2
**Rating:** 4
**Confidence:** 4

**Summary:**

The paper proposes DualMap, a dual-mapping scheduling framework for distributed LLM serving that reconciles the trade-off between cache affinity and load balancing. By mapping each request to two candidate instances via independent hash functions and using SLO-aware routing, hotspot-aware rebalancing, and dual-hash-ring scaling, DualMap achieves both efficient KV cache reuse and balanced workloads. Experiments on real-world traces show up to 2.25× higher request capacity compared to state-of-the-art schedulers.

**Strengths:**

1. The problem is well defined: the trade-off between cache affinity and load balancing in LLM serving.
2. Extends the “power of two choices” concept to LLM scheduling, offering a novel way to achieve both objectives simultaneously.
3. The evaluation id comprehensive. Benchmarks across models and different baselines clearly show the superior performance.

**Weaknesses:**

1. The motivation for using two hashes for scheduling is unclear. Is it to save scheduling latency? Or, why not collect global information from all workers and then choose the best one (e.g. based on a weighted sum of prefix-cache and balance benefits)? The paper is very unclear on this point.
2. While “power of two choices” is cited, formal analysis of DualMap’s convergence or optimality is limited.
3. Lacks of scheduling overhead analysis.

**Questions:**

1. Is there any challenge integrating Dualmap into disaggregation serving system (e.g. Prefill-Decode disaggregation, Attention-FFN disaggregation)?
2. Could you please further explain the potential drawbacks of using global information for scheduling compared to Dualmap?

---

> ### Author Response · Authors · 2025-11-21
> **Official Comment by Authors**
>
> We thank the reviewer for carefully reading our paper and for the constructive suggestions on theory, motivation, overhead, and integration with disaggregated architectures. We have revised the paper accordingly and summarize the key changes below.
>
> ### **W1 & Q2: Why Two Hashes? Why Not Global Information?**
>
> We appreciate you noting that the motivation for using two hashes was unclear. We have revised §2.3 to clarify:
>
> DualMap uses two hashes because it simultaneously provides (i) *provable load balancing guarantees* and (ii) *high cache locality*. This follows the classical PoTC paradigm. Consider $m$ requests distributed over $n$ instances, each request with $d$ randomly selected candidates. As discussed in **W2**, the choice of $d$ critically affects load balancing. Compared with single-choice scheduling ($d=1$), which yields a maximum load deviation of
>
> $$
> \Theta\Big(\sqrt{\frac{m \log n}{n}}\Big),
> $$
>
> using two choices ($d=2$) reduces the maximum load deviation to at most an exponential improvement.
>
> $$
> \frac{\log \log n}{\log 2} + \mathcal{O}(1),
> $$
>
> Increasing $d$ beyond 2 provides only marginal gains while degrading prefix locality, as requests sharing the same prefix are spread across more instances.
>
> A global strategy that “collects information from all workers and picks the best one” is effectively equivalent to $d = n$. Theory shows this yields little improvement in maximum load compared to $d=2$, but significantly harms cache affinity by scattering identical prefixes across the cluster, increasing prefill computation cost and TTFT.
>
> ### **W2: Limited Formal Analysis**
>
> Thank you for noting the limited formal analysis. We have revised §2.3 accordingly.
>
> DualMap follows PoTC, providing strong theoretical guarantees. Consider $m$ requests sent to $n$ instances, each with $d$ randomly selected candidates.
>
> **1. Single-choice ($d=1$).**
>
> The maximum load deviates significantly from the average:
> $$
> \max_i L(I_i) = \frac{m}{n} + \Theta\Big(\sqrt{\frac{m \log n}{n}}\Big)
> \tag{1}
> $$
>
> where:
>
> - $L(I_i)$ : the number of requests (load) assigned to instance $I_i$.
> - $\frac{m}{n}$ :the *average load per instance*, i.e., the ideal load if $m$ requests were balanced across $n$ instances.
> - $\Theta\Big(\sqrt{\frac{m \log n}{n}}\Big)$ : the *deviation from the average load* due to random assignment. It grows with $m$ and $n$, reflecting the imbalance.
>
> **2. Two-choice ($d=2$).**
>
> For $d \ge 2$, PoTC guarantees that the maximum load satisfies:
> $$
> \max_i L(I_i) \le \frac{m}{n} + \frac{\log \log n}{\log d} + \mathcal{O}(1)
> \tag{2}
> $$
>
> $\frac{\log \log n}{\log d}$captures deviation due to random assignment. For $d=2$, it becomes $\log \log n$, which is much smaller than $\Theta\Big(\sqrt{\frac{m \log n}{n}}\Big)$ for $d=1$. Compared to $d=1$, $d=2$ provides exponentially better load balancing.
>
> **3. More than two choices ($d>2$).**
> Increasing $d$ beyond two yields diminishing returns. The maximum deviation  $\frac{\log \log n}{\log d}$ for $d = 2, 3, 4$:
>
> - **d = 2**: $\log \log n$
>
> - **d = 3**: $0.91 \log \log n$
>
> - **d = 4**: $0.72 \log \log n$
>
> Going from $2 \to 3 \to 4$ choices provides only marginal gains.  Thus, *DualMap* uses $d=2$ to achieve load balancing while ensuring that each request is scheduled only among two candidates. This preserves cache affinity, as requests sharing the same prefix are mapped to the same pair of instances.
>
>
> ### **W3: Scheduling Overhead Analysis**
>
> We added a quantitative overhead evaluation in the revision (Appendix A.4).
>
> * *Metadata overhead.* For Qwen2.5-7B, a 64 GB KV cache corresponds to 9,632 blocks; with 16 B of metadata per block, this is ≈146.2 KB per instance and ≈4.57 MB for 32 instances. For Qwen2.5-14B, the per-instance and 32-instance footprints are ≈44 KB and ≈1.38 MB.
>
> * *Runtime overhead.* KV-cache metadata access ≈0.2 ms per request; dual-hash + TTFT estimation ≈0.6 ms per request; hotspot-aware rebalancing 2.2–2.5 ms per invocation, scaling with the overloaded queue length but not with cluster size. Compared to typical prefill latencies, these costs are negligible.
>
> These demonstrate that DualMap’s overhead is small and well-bounded.
>
>
> ### **Q1：Integration into Disaggregated Architectures**
>
> We appreciate this forward-looking question and have added a discussion in Appendix A.6.  DualMap is agnostic to internal micro-architecture, requiring only estimated TTFT per candidate.
>
> - *Prefill–Decode Disaggregation:*  TTFT is dominated by prefill; DualMap routes prefill instances, while decode nodes attach according to the existing framework, requiring no invasive changes.
>
> - *Attention–FFN Disaggregation*: AF disaggregation optimizes the decode stage.  DualMap focuses on scheduling Prefill (P) instances; since AF disaggregation is downstream, it does not affect DualMap’s TTFT-based routing  and rebalancing.
>
> Thank you again for your thoughtful comments. We appreciate your time and remain available to answer any further questions.

---

> ### Author Response · Authors · 2025-11-28
>
> Dear Reviewer 6DK7,
>
> We hope this message finds you well. As the discussion period is nearing its end, we want to ensure that we have addressed all your concerns satisfactorily. If there are any additional points or feedback you would like us to consider, please let us know. Your insights are invaluable to us, and we are eager to address any remaining issues to further improve our work.
>
> Thank you very much for your time and effort in reviewing our paper.
>
> Best regards,
>
> The Authors

---

### Official Review · Reviewer_iBGX · 2025-11-01

**Soundness:** 4
**Presentation:** 3
**Contribution:** 4
**Rating:** 8
**Confidence:** 3

**Summary:**

This paper presents DualMap, a scheduling strategy for distributed LLM serving that considers both the cache affinity and load balancing. DualMap prepares two independent hash functions and selects the better one. It implements SLO-aware request routing, hotspot-aware rebalancing, and lightweight dual-hash-ring scaling. DualMap improves the effective request capacity by up to 2.25x.

**Strengths:**

- This paper solves the real trade-off in the LLM serving system: load balancing vs. cache affinity.
- Comprehensive evaluation to show the advantage of the proposed method against multiple baselines.

**Weaknesses:**

- There are a few points that are unclear to me. See questions.

**Questions:**

- What is the traffic ratio ρ? It seems the paper lacks definition.
- What happens if two hash maps return the same instance?
- How do you choose the instance to migrate ($j$ in Equation 3)? Do you search over all possible $j$?
- In section 3.4, this part was not clear to me: "Because request-to-instance mappings are determined by relative positions on the ring, changes to cluster membership affect only localized regions. This design ensures that most requests retain their original mapping paths during scaling operations"
- I am curious if two is the optimal number for hash maps. Would there be any benefits or disadvantages in having three or more hash maps?

---

> ### Author Response · Authors · 2025-11-21
> **Official Comment by Authors**
>
> Thank you for the thoughtful questions and for recognizing the contribution of DualMap. Below we address each point and describe the corresponding clarifications added to the paper.
>
> ### **Q1: Definition of Traffic Ratio $\rho$**
>
> You are right that the original version did not explicitly define ρ. We now define it in §3.2: $\rho$ is the traffic ratio of a prefix, i.e., the fraction of total requests that share the same prefix within a given time window.
>
> We use $\rho$ to quantify prefix “hotness”; a larger $\rho$ means more requests share that prefix. A prefix is considered hot when $\rho$ > 2/n, where n is the number of instances, reflecting the upper bound induced by DualMap’s dual-mapping strategy.
>
>
>
> ### **Q2: What if the Two Hashes Return the Same Instance? How Is j Chosen in Eq. (3)?**
>
> DualMap is designed so that every request has two distinct candidate instances. If the two hash functions $f_1$, $f_2$ initially map a request to the same instance, we deterministically adjust the second candidate as
>
>
> $$
> instanceId_2 = (instanceId_1 + 1) \bmod numInstances
> $$
>
>
>
>
> This ensures that each request always has two distinct candidate instances. We have clarified this in §3.2.
>
> For hotspot-aware rebalancing (originally Eq.(3), revised as Eq.(6)), the alternative instance *j* is exactly this second candidate from the dual mapping. We do not search over all instances; instead, we preserve the prefix-bound candidate pair {I1, I2} and only migrate within the pair. This keeps cache affinity and scheduling complexity under control. We have clarified this in §3.3.
>
> ### **Q3: Clarification on Localized Impact During Elastic Scaling**
>
> We have rewritten this part in Appendix A.2.3 to be more concrete. In DualMap, all instances and request prefixes are placed on a logical ring using their hash values. Each hashed prefix picks the nearest clockwise instance as its candidate. Thus, mappings depend only on *relative positions* on the ring.
>
> Adding an instance inserts a new anchor point on the ring. Only requests whose hashed positions fall between the previous and the new anchor change their mapping; other requests keep their original instances. Removing an instance similarly only affects prefixes whose nearest clockwise instance was that node.
>
> We now include a small textual example in the paper to illustrate how adding a new instance between B and C on the ring only remaps requests in (B, E], while all others remain unchanged.
>
> **Simplified illustration:**
>
> Logical ring before adding E:
>
>   ... --- (A) --- (B) ---------------- (C) --------------- (D) --- ...
>
> Requests hashed between:
>
>  - (B, C) -> map to C
>  - (C, D) -> map to D
>  - (A, B) -> map to B
>
> Logical ring after adding E between B and C:
>
>   ... --- (A) --- (B) --- (E) ---------- (C) --------------- (D) --- ...
>
> Now only requests hashed in (B, E] change mapping:
>
>  - (B, E] -> map to E   <-- newly remapped
>  - (E, C] -> still map to C
>  - others -> unchanged
>
> ### **Q4: Is Two the Optimal Number of Hash Maps? What About Three or More?**
>
> DualMap instantiates the classical Power-of-Two-Choices (PoTC) result. For $m$ requests and $n$ instances, the maximum load under $d$ choices satisfies [1] :
> $$
> \max_{i} L(I_i) \le \frac{m}{n} + \frac{\log \log n}{\log d} + \mathcal{O}(1)
> $$
>
> where the first term $\frac{m}{n}$ is the average load and the second term $\frac{\log \log n}{\log d}$ captures deviation due to the randomness of the $d$-choice allocation.
>
> - For $d=2$, this reduces to the classical PoTC result:
>
> $$
> \max_{i} L(I_i) \le \frac{m}{n} + \log \log n + \mathcal{O}(1)
> $$
>
> which is used by *DualMap*.
>
> - For $d>2$, the maximum load slightly decreases, but the benefit exhibits *diminishing returns*.  Maximum deviation $\frac{\log \log n}{\log d}$ under different $d$:
>
>   - **d = 2**: $\log \log n$
>
>   - **d = 3**: $0.91 \log \log n$
>
>   - **d = 4**: $0.72 \log \log n$
>
> As shown, moving from 2 → 3 → 4 choices provides only marginal improvements in load balancing. Additionally, using more than two choices can reduce cache affinity, as requests with the same prefix are more likely to be distributed across multiple instances, lowering KV cache reuse. Therefore, two choices strike a practical balance between strong load balancing and high cache affinity.
>
> We appreciate your comments; the definitions and explanations added to the revised paper in §2.3 directly address your questions and make the design of DualMap more transparent.
>
>
>
> ### **References**
>
> [1] Mitzenmacher, M. (2002). The power of two choices in randomized load balancing. *IEEE Transactions on Parallel and Distributed Systems*, *12*(10), 1094-1104.

---

> > ### Comment · Reviewer_iBGX · 2025-11-27
> >
> > Thanks so much for the response. That clarifies all of my questions.

---

### Official Review · Reviewer_JVha · 2025-11-10

**Soundness:** 4
**Presentation:** 4
**Contribution:** 4
**Rating:** 8
**Confidence:** 4

**Summary:**

DualMap build a system to better optimally allocate resources balancing both cache load and cache affinity. They describe ways to determine the appropriate hash length, SLO aware scheduling based on the violation of TTFT, and a rebalancing.

**Strengths:**

I found this paper to be a good extension on existing work and provide decent/extensive results.

They contribute and interesting hotspot aware rebalancing and light weight rebalancing.

**Weaknesses:**

There seems to be a lack of scalability/scheduler overhead analysis in the implementation. It would be interesting to see on more GPUs(even if simulated).

The workload talks about cache migration based on TTFT but it would also be interesting if a direct NVLink transfer/memory cache state awareness was added to this policy.

**Questions:**

1. Possibly a scalability/scheduler analysis could be good

---

> ### Author Response · Authors · 2025-11-21
> **Official Comment by Authors**
>
> Thank you for the positive assessment of DualMap and for pointing out the missing scalability and scheduler-overhead analysis. We have substantially extended the paper to address these concerns.
>
> **Scalability Analysis**. In the revised manuscript, we add a new experiment using a Vidur-based simulator [1] to evaluate DualMap at larger scales. We simulate a cluster of instances, each with 64 GB DRAM and one NPU, serving Qwen2.5-7B on the Tool&Agent workload. The number of instances is scaled from 8 to 32, and the total number of requests is increased proportionally from 8K to 32K.
>
> As shown in the table below (now in Appendix A.4), DualMap achieves near-linear growth in goodput (maximum sustainable request rate under the 90% TTFT SLO) and consistently outperforms Cache Affinity, Least Loaded, Min TTFT, and Preble across all cluster sizes:
>
> | Scheduler      | 8 instances | 16 instances | 24 instances | 32 instances |
> | -------------- | ----------- | ------------ | ------------ | ------------ |
> | Cache Affinity | 4.25        | 8.5          | 12.5         | 17.2         |
> | Least Loaded   | 5.15        | 10.5         | 16           | 21.5         |
> | Min TTFT       | 6.15        | 12.5         | 18           | 26           |
> | Preble         | 3.5         | 6.3          | 6.4          | 6.7          |
> | **DualMap**    | 6.4         | 13.5         | 21.2         | 28.7         |
>
> These results indicate that DualMap’s dual-mapping design scales well: prefix-bound dual hashing preserves cache affinity, while rebalancing always operates within candidate pairs, keeping the load-balancing behavior stable as the cluster grows.
>
> **Scheduler Overhead**. We also add a quantitative overhead analysis:
>
> * *Metadata footprint.* Following Mooncake/APC-style[2] designs, each 128-token block stores a hash value and block ID (8 B each), i.e., 16 B per block. For Qwen2.5-7B, a 64 GB KV cache contains 9,632 blocks, so each instance stores about 146.2 KB of metadata; a 32-instance cluster stores 4.57 MB. For Qwen2.5-14B, the per-instance and 32-instance metadata costs are 44 KB and 1.38 MB, respectively, growing linearly with total KV-cache capacity.
>
> * *Runtime footprint.* On our testbed, the per-request runtime overhead is also bounded and independent of cluster size. KV-cache query and save take ~0.2 ms per request, SLO-aware routing takes ~0.6 ms, and hotspot-aware rebalancing takes 2.2–2.5 ms per invocation.
>
> Compared to prefill-stage latencies on the order of seconds, these costs are negligible. Detailed experimental settings and numbers are now reported in Appendix A.4.
>
> We thank you again for your thoughtful feedback. The added scalability experiments and overhead analysis substantially strengthen the paper along the exact dimensions you raised. We hope these revisions will positively inform your final assessment.
>
>
>
> **References**
>
> [1] Amey Agrawal, Nitin Kedia, Jayashree Mohan, Ashish Panwar, Nipun Kwatra, Bhargav S Gulavani,Ramachandran Ramjee, and Alexey Tumanov. Vidur: A large-scale simulation framework for llm inference. Proceedings of Machine Learning and Systems, 6:351–366, 2024.
>
> [2] https://github.com/vllm-project/vllm/issues/2614

---

### Author Response · Authors · 2025-12-01
**Summary Comment for Paper 11519 (1)**

We thank all reviewers for their thoughtful feedback and constructive suggestions. We have carefully revised the paper and substantially strengthened both the technical clarity and empirical depth of our work. Below, we summarize the major revisions made in response to each reviewer’s concerns.

---

### **1. Reviewer JVha (Initial Rating: 8)**

#### **Key Concerns**

- Lack of scalability and overhead analysis

#### **Our Responses & Revisions**

- **Scalability Analysis:**
  We added Vidur-based simulation experiments (Appendix A.4.1), scaling the system from 8 to 32 instances with a proportionally increased request volume. Results demonstrate that DualMap achieves near-linear goodput scaling and consistent performance improvements over baseline methods (Cache Affinity, Least Loaded, Min TTFT, and Preble).
- **Scheduler-Overhead Analysis:**
  We added quantitative evaluation of metadata and runtime overhead (Appendix A.4.2).
  - Metadata footprint grows linearly with KV-cache capacity  (146 KB per instance for Qwen2.5-7B; 44 KB for Qwen2.5-14B).
  - Runtime overhead per request remains small and independent of cluster size  (KV-cache query/save: ≈0.2 ms; SLO-aware request routing: ≈0.6 ms; hotspot-aware request rebalancing: 2.2–2.5 ms per invocation).

#### **Current Status**

Although we have not yet received any follow-up from reviewer JVha, we believe all concerns have been thoroughly addressed.

---

### **2. Reviewer iBGX (Initial Rating: 8)**

#### **Key Concerns**

- Definition of the traffic ratio ρ unclear
- What if the two hash functions map to the same instance?
- Localized impact during elastic scaling unclear
- Why choose two hash maps instead of three or more?

#### **Our Responses & Revisions**

- **Definition of Traffic Ratio ρ:**

  We now formally define ρ as the fraction of requests sharing a prefix within a time window, and explicitly specify this definition in §3.2.

- **Two Hashes Returning the Same Instance:**
  If both hashes select the same instance, the second candidate is deterministically shifted to a logically adjacent instance to ensure two distinct choices (§3.2). During rebalancing, request migration is restricted to the request’s two candidate instances. (§3.3).

- **Localized Impact During Elastic Scaling:**
  Clarified that DualMap places all instances and prefixes on a logical ring using hash values; each prefix selects the nearest clockwise instance. Mappings depend solely on relative ring positions (Appendix A.2.3).

- **Why Two Hash Maps (and Not More)?**
  PoTC analysis shows that increasing the number of choices from 2 to 3 to 4 yields only marginal load-balancing gains, while using more than 2 choices significantly impairs cache affinity by spreading shared prefixes across multiple instances. Two hash maps thus offer the optimal balance between load balancing and cache affinity (§2.3).

#### **Current Status**

Reviewer iBGX responded that our clarifications have resolved all their concerns.

---

### 3. **Reviewer 6DK7 (Initial Rating: 4)**

#### **Key Concerns**

- Why two hashes? Why not use global information?
- Limited formal analysis of DualMap
- Missing scheduler-overhead evaluation
- Integration with disaggregated architectures (Prefill–Decode, Attention–FFN)

#### **Our Responses & Revisions**

- **Why Two Hashes / Why Not Global Information?**
  This question overlaps with reviewer iBGX’s concerns, and our explanation—now endorsed by reviewer iBGX—is incorporated into §2.3 and Appendix A.9.

- **Formal Analysis:**
  We added PoTC-based analysis demonstrating that two choices are sufficient for load balancing while preserving strong cache affinity by constraining candidate instances (§2.3, Appendix A.9). Reviewer J9qz also endorsed this explanation in cross-review.

- **Scheduler Overhead:**

  This concern has been addressed using the same metadata and runtime evaluations added in response to reviewer JVha (Appendix A.4).

- **Integration with Disaggregated Architectures:**
  We discussed how DualMap schedules only Prefill (P) instances and requires no intrusive modifications to Prefill–Decode or Attention–FFN disaggregated systems (Appendix A.6).

#### **Current Status**

Although reviewer 6DK7 has not yet replied, we believe all issues have been fully addressed.

---

> ### Author Response · Authors · 2025-12-01
> **Summary Comment for Paper 11519 (2)**
>
> ### **4. Reviewer J9qz (Initial Rating: 6)**
>
> #### **Key Concerns**
>
> - Novelty relative to prior systems (e.g., Dynamo, Chord, consistent hashing)
> - TTFT estimation may be oversimplified in mixed workloads and non-FCFS scheduling
> - Additional clarity on KV cache prefix-sharing statistics and Figures 1, 3, and 7
> - Suggested adding results visualizing why two hashes are optimal
>
> #### **Our Responses & Revisions**
>
> - **Clarifying Novelty and Contribution:**
>   We highlighted LLM-specific constraints (prefix locality + TTFT SLO) and our design contributions—prefix-based dual-mapping, SLO-aware request routing, and hotspot-aware request rebalancing. These jointly achieve both cache affinity and load balancing, which traditional systems cannot (Appendix A.7).
>
> - **TTFT Estimation Robustness:**
>   - Under vLLM’s prefill-priority scheduling,  a prefill request waits at most for a single decode batch (milliseconds), negligible compared to the waiting time for other prefill requests, typically on the order of seconds.
>   - Memory contention resembles an overloaded-instance state; our hotspot-aware request rebalancing mechanism (§3.3) was designed from the outset not only to rebalance general overloaded instances but also to resolve this scenario.  We have added a detailed explanation in Appendix A.8 of the revised version.
>   - We discussed a future extension that introduces a global FCFS queue to ensure reliable TTFT estimation, regardless of backend local scheduler policies.
>
> - **Figures & Prefix Statistics:**
>   We added new CDF plots quantifying shared-prefix rates in real-world scenarios (Appendix A.5) and replotted Figures 1, 3, and 7 in accordance with the reviewers’ suggestions.
>
> - **Two Hashes Are Optimal – Visualization:**
>   We added a dedicated plot in Appendix A.9 demonstrating the effectiveness of two choices in achieving cluster load balancing.
>
> #### **Current Status**
>
> Reviewer J9qz acknowledged that the paper is now substantially stronger in terms of novelty and clarity, and agreed that our simplification of TTFT estimation is reasonable. Our second response addresses the remaining question regarding TTFT estimation under memory contention.
>
> ---
>
> ### **Summary**
>
> Through substantial theoretical additions (PoTC analysis), expanded experimental evaluations (scalability, overhead), and enhanced design clarity (robustness of TTFT estimation, novelty of  DualMap), we are confident that all reviewers’ core concerns have been thoroughly addressed. We thank all reviewers for their constructive input, which has meaningfully improved the rigor and clarity of the paper.
>
> Best regards,
>
> The authors of paper 11519

---

### Meta-Review · Area_Chair_q5ix · 2026-01-06

**Summary:**

The reviewers agree that this paper makes a strong and practical contribution to distributed LLM serving by clearly formulating and effectively resolving the long-standing tension between cache affinity and load balancing through the DualMap scheduling framework. By combining prefix-aware dual hashing with SLO-aware routing and hotspot-aware rebalancing, the paper demonstrates substantial improvements in effective request capacity under realistic TTFT constraints. Reviewers found the system design well-motivated, the empirical evaluation thorough across multiple baselines and workloads, and the integration of classical power-of-two-choices theory with LLM-specific constraints both principled and impactful.

**Reviewer Concerns:**

Initial concerns regarding novelty relative to prior hashing-based systems, the choice of two hash functions, TTFT estimation robustness, scalability, and scheduler overhead were comprehensively addressed in the rebuttal through added theoretical analysis, expanded large-scale and overhead experiments, clearer positioning against prior systems, and improved presentation and workload characterization.

**Reviewer Scores:**

After discussion and rebuttal, reviewers converged toward positive assessments, with multiple reviewers at clear accept scores and initially borderline reviewers indicating that their concerns had been resolved and that the paper is now in strong shape for acceptance.

---

### Decision · Program_Chairs · 2026-01-26

Accept (Poster)